# Shared genetic architectures of educational attainment in East Asian and European populations

Tzu-Ting Chen [1,26], Jaeyoung Kim [2,3,26], Max Lam [4,5,6,7,8], Yi-Fang Chuang[9], Yen-Ling Chiu[10,11], Shu-Chin Lin[1], Sang-Hyuk Jung [12], Beomsu Kim[2], Soyeon Kim [4,13], Chamlee Cho [2], Injeong Shim [2], Sanghyeon Park [2], Yeeun Ahn [2], Aysu Okbay [14], Hyemin Jang[15,16], Hee Jin Kim[15,16], Sang Won Seo[15,16], Woong-Yang Park[17], Tian Ge[13,18,19], Hailiang Huang [4,5,13], Yen-Chen Anne Feng [20,21], Yen-Feng Lin [1,22,23,27] ✉, Woojae Myung [3,24,27] ✉, Chia-Yen Chen [25,27] ✉ & Hong-Hee Won [2,17,27] ✉

Educational attainment (EduYears), a heritable trait often used as a proxy for cognitive ability, is associated with various health and social outcomes. Previous genome-wide association studies (GWASs) on EduYears have been focused on samples of European (EUR) genetic ancestries. Here we present the first large-scale GWAS of EduYears in people of East Asian (EAS) ancestry ($n$ = 176,400) and conduct a cross-ancestry meta-analysis with EduYears GWAS in people of EUR ancestry ($n$ = 766,345). EduYears showed a high genetic correlation and power-adjusted transferability ratio between EAS and EUR. We also found similar functional enrichment, gene expression enrichment and cross-trait genetic correlations between two populations. Cross-ancestry fine-mapping identified refined credible sets with a higher posterior inclusion probability than single population fine-mapping. Polygenic prediction analysis in four independent EAS and EUR cohorts demonstrated transferability between populations. Our study supports the need for further research on diverse ancestries to increase our understanding of the genetic basis of educational attainment.

Educational attainment (EduYears: years of education as a continuous phenotype) is a behavioural trait that has been studied extensively and linked to various social, economic and health-related outcomes[1–3]. While EduYears is an important trait studied in behavioural genetics, it has also been a topic of interest in epidemiology and medical research owing to its phenotypic and genetic correlation with various diseases, including cardiovascular diseases[4], metabolic diseases[5], psychiatric disorders[6], Alzheimer's disease[7] and cognitive function[8]. As educational attainment can be easily measured through self-report within large population samples (as opposed to disease status such as schizophrenia and Alzheimer's disease), it is considered a useful proxy phenotype for characterizing related health outcomes[9,10].

EduYears has been shown to be moderately heritable with a heritability of ~40% from twin studies and a single-nucleotide polymorphism (SNP)-based heritability of ~20% from genome-wide association studies (GWASs)[11]. The genetic study of EduYears thus offers insights into the factors contributing to its observed variation across populations. Previous GWAS meta-analyses and replication studies have identified genetic variants associated with EduYears[11–14]. Notably, the largest GWAS meta-analysis for EduYears, which included approximately 3 million individuals of European (EUR) genetic ancestries, has identified 3,952 independent genome-wide significant loci[10]. However, previous studies investigated the genetic architecture of EduYears solely focused on samples of EUR genetic ancestries, and the evidence

for the generalizability of the findings to non-EUR populations is limited. The lack of diversity in genetic studies on EduYears could lead to social and health disparities due to inadequate comprehension of EduYears and its impact on socioeconomic and health outcomes in understudied populations[15].

In this Article, we conducted the first large-scale EduYears GWAS in the East Asian (EAS) population, followed by a cross-ancestry GWAS meta-analysis for EduYears between EAS and EUR populations. The primary objectives of this study were to (1) identify genomic loci for EduYears in cross-population samples, (2) investigate the biological basis of EduYears in the EAS population, (3) examine whether the genetic architecture of EduYears is shared between EAS and EUR populations, and (4) demonstrate the advantages of cross-population analysis in polygenic prediction and fine-mapping of causal variants. With public sharing our summary results, our findings will facilitate future studies on diverse genetic ancestries and enhance our knowledge of the genetic basis for educational attainment.

## Results

### EduYears genome-wide associations in EAS population

Self-reported educational attainment (EduYears) and genome-wide genotype data for 107,493 and 72,294 samples were obtained from the Taiwan Biobank (TWB)[16] and Korean Genome and Epidemiology Study (KoGES)[17], respectively. After stringent quality control (QC) and genotype imputation, we performed a GWAS for EduYears with 7,470,871 variants in 104,722 TWB samples and 8,064,004 variants in 71,678 KoGES samples (Supplementary Figs. 1a,b and 2a,b and Supplementary Table 1). We then performed an EAS genome-wide fixed-effect meta-analysis for EduYears between TWB and KoGES, which retained the association results for 6,951,085 autosomal variants with an imputation quality score (INFO) >0.6 and minor allele frequency (MAF) >0.5% in both cohorts (Fig. 1a, Supplementary Fig. 1c and Supplementary Table 1). The results of the EAS GWAS meta-analysis were similar to those of the TWB and KoGES separately, except for one locus on chromosome 12 near *ALDH2* that showed significant heterogeneity (Supplementary Fig. 3). Genome-wide associations in the TWB, KoGES and their meta-analysis were consistent with the highly polygenic architecture of educational attainment and did not indicate inflation due to potential population stratification (linkage disequilibrium score regression (LDSC)[18] intercept ranged from 1.029 to 1.046). The $\lambda_{GC}$ ranged from 1.165 to 1.320 in EAS and was 2.094 for EUR based on publicly available data[14] ($n$ = 766,345) and 2.807 for EUR as reported by Lee et al.[14] ($n$ = 1,131,881; Supplementary Figs. 1a–c and Supplementary Table 2). In total, we identified seven genome-wide significant loci ($P < 5 \times 10^{-8}$), including 11 independent SNPs, from the GWAS meta-analysis for EduYears in EAS (Fig. 1a, Table 1 and Supplementary Table 3). All these 11 independent SNPs were previously reported (SNPs located within ±500 kb of the 3,952 lead SNPs reported by Okbay et al.[10]).

We used several approaches to examine the consistency of genetic effects for EduYears between the TWB and KoGES. At the genome-wide level, we first showed that the SNP-based heritability for EduYears was estimated to be 9.7% in TWB, 8.7% in KoGES and 9.0% in the EAS GWAS meta-analysis, and the genome-wide genetic correlation for EduYears between TWB and KoGES was 0.871 (standard error (s.e.) 0.073) using LDSC[18] (Fig. 2a and Supplementary Table 2). In addition, the mean of genome-wide fixation index (Fst) between TWB and KoGES was 0.005, which suggests small population differences due to genetic background (Supplementary Table 4). At the individual locus level, we observed that the direction of genetic effects was consistent between TWB and KoGES for most genome-wide significant SNPs, except for the genome-wide significant locus on chromosome 12 (Supplementary Figs. 4a,b and 5a–g and Supplementary Table 3). The effect allele frequency also showed high consistency between TWB and KoGES for the variants included in the GWAS meta-analysis in the EAS population (Supplementary Fig. 6).

### Heterogeneity of genetic effects within EAS population

Given that the *ALDH2* region on chromosome 12 showed a significant association with EduYears exclusively in KoGES but not in TWB, we conducted further investigation to explore potential underlying factors driving this observed heterogeneity. Firstly, we examined the phenome-wide association study results for the *ALDH2* region in KoGES and demonstrated that total alcohol consumption exhibited the most significant association with this locus[19] (Supplementary Table 5). Based on this finding, we estimated the genetic correlation ($r_g$) between alcohol drinking and EduYears in KoGES, both globally and locally. We identified a significant negative global genetic correlation between alcohol drinking and EduYears ($r_g$ = −0.193; s.e. 0.063; $P$ = 0.002). Moreover, specifically within the *ALDH2* region, we observed a substantial local genetic correlation ($\rho$ = −0.82, $P$ = 7.4 × 10⁻⁶). In addition, we conducted a stratified GWAS for EduYears, segregating KoGES participants into groups of drinkers and non-drinkers. Remarkably, in the drinker group, the *ALDH2* region displayed a significant association with EduYears ($P$ = 2.4 × 10⁻²²), while in the non-drinker group, the association was not significant ($P$ = 0.032) (Supplementary Fig. 7). These findings suggest that the observed heterogeneity in the *ALDH2* region is probably attributed to potential shared genetic component and gene–environment interactions between alcohol drinking and EduYears, particularly in KoGES.

### Potential biological mechanisms underlying EduYears in EAS

To elucidate the underlying biological mechanisms of EduYears in the EAS population, we first applied expression quantitative trait loci (eQTL) mapping and MAGMA gene-set analysis[20,21] implemented in FUMA v1.3.7 (ref. 22) to identify potentially causal genes and gene sets. For eQTL mapping, we identified 13 genes mapped to the EAS EduYears loci through *cis*-eQTL using 13 brain tissue types from the Genotype-Tissue Expression (GTEx) v8 dataset[23] (Supplementary Table 6). Notably, the lead SNP rs12936234 was mapped to three genes, namely *DCAKD*, *NMT1* and *C1QL1* in ten brain tissues. We did not identify any significant gene-set association after multiple comparison correction, while the amyloid β metabolic process was the most significant Gene Ontology pathway ($P$ = 3.59 × 10⁻⁵; Supplementary Table 7).

Second, we employed a stratified LDSC[24,25] with 97 baseline linkage disequilibrium (LD) annotations[26] for our EAS GWAS summary statistics and EUR summary statistics by Lee et al.[14]. Among the 97 stratified LDSC annotations, we observed significant enrichments for EduYears in the EAS population in six annotations, including H3K4me1 peaks (false discovery rate (FDR) <5%; Supplementary Fig. 8 and Supplementary Table 8). In the EUR population, 17 annotations, including the conserved primate phastCons46way annotation, representing genomic regions conserved across primate species, showed significant enrichment for EduYears (FDR <5%; Supplementary Table 9). Furthermore, ten MAF binary annotations were included to model MAF-dependent architectures within the set of 97 annotations. Of these ten MAF bins, five (more common MAF bins) exhibited significant enrichments for EduYears in both EAS and EUR populations.

Third, to determine the tissues and cell types associated with EduYears, we conducted LDSC applied to specifically expressed genes (LDSC-SEG) analysis[25]. For analysis across multiple tissues, we used gene expression data from the GTEx, Franke laboratory and Cahoy et al. (see Uniform Resource Locators (URLs)). In the EAS population, EduYears-associated SNPs were strongly enriched in the brain, parietal lobe and putamen of the central nervous system at an FDR <5% threshold (Supplementary Fig. 9a and Supplementary Table 10), which is consistent with previously published LDSC-SEG results in EUR[14]. We also used chromatin data from the Roadmap Epigenomics and ENCODE projects for the LDSC-SEG analysis. In the EAS population, SNP heritability was significantly enriched in the central nervous system, including the foetal brain, dorsolateral prefrontal cortex and inferior

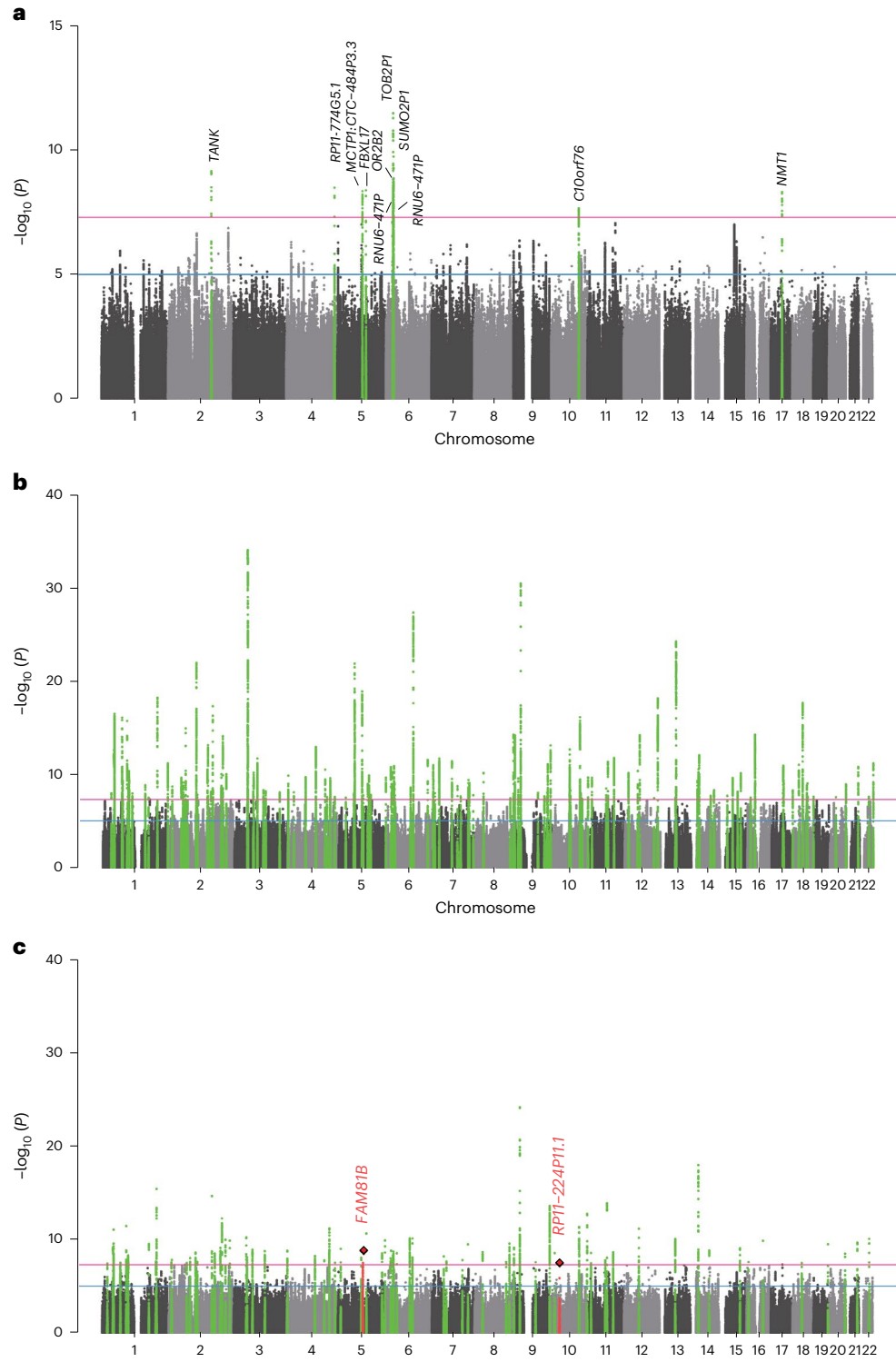

**Fig. 1 | Gene discovery through common variant associations for EduYears in EAS and EUR populations. a**, Manhattan plot of genome-wide meta-analysis for EduYears in EAS. **b**, Manhattan plot of cross-population genome-wide meta-analysis for EduYears in EAS and EUR. **c**, Manhattan plot of MAMA for EduYears in EAS. The x axis represents chromosomal position, and the y axis represents the $-\log_{10}(P$ value) for the association of variants with EduYears. Reported P values are two-sided and not corrected for multiple testing. Independent SNPs are highlighted in green, and previously unreported SNPs are highlighted with a red diamond. The horizontal pink line marks the threshold for genome-wide significance ($P = 5 \times 10^{-8}$), and the horizontal blue line marks the threshold for suggestive genome-wide significance ($P = 1 \times 10^{-5}$).

temporal lobe after FDR correction (Supplementary Fig. 9b and Supplementary Table 11), which is also consistent with previous results in EUR[14]. By using the Cahoy dataset[27] to examine the enrichment in three brain cell types (neurons, astrocytes and oligodendrocytes), we identified that EduYears-associated SNPs were more enriched in neurons than astrocytes or oligodendrocytes in the EAS population (Supplementary Table 12). The enrichment in neurons was also found in previous EUR studies[10,14].

**Table 1 | Genome-wide significant loci for EduYears in EAS population**

| Chr | Start | End | Credible set ID | Credible set size | Total PIP | *N* of significant variants[a] | SNP with maximum PIP | Maximum PIP | A1 | Effect size | A1 Freq | Marginal *P* value | Gene | Annotation |
|---|---|---|---|---|---|---|---|---|---|---|---|---|---|---|
| 2 | 161,721,597 | 162,351,261 | 1 | 6 | 0.96 | 6 | rs10930013 | 0.26 | A | −0.021 | 0.42 | 7.80×10⁻¹⁰ | *TANK* | Intron |
| 4 | 180,613,679 | 181,136,169 | 1 | 4 | 0.95 | 4 | rs2871133 | 0.36 | C | −0.020 | 0.53 | 3.54×10⁻⁹ | NA | NA |
| 5 | 93,838,858 | 94,392,030 | 1 | 20 | 0.95 | 19 | rs255347 | 0.11 | T | 0.022 | 0.73 | 5.70×10⁻⁹ | *MCTP1* | Intron |
| 5 | 106,947,725 | 107,455,182 | 1 | 2 | 0.96 | 2 | rs7708343 | 0.62 | A | −0.021 | 0.32 | 4.46×10⁻⁹ | *FBXL17* | Intron |
| 6 | 27,515,505 | 29,611,229 | 1 | 13 | 0.95 | 3 | rs9461540 | 0.47 | G | −0.027 | 0.17 | 6.19×10⁻¹⁰ | *GABBR1* | Upstream gene |
| 6 | 27,515,505 | 29,611,229 | 2 | 6 | 0.96 | 6 | rs16893804 | 0.40 | G | −0.028 | 0.22 | 5.62×10⁻¹² | NA | NA |
| 10 | 103,385,878 | 104,057,295 | 1 | 27 | 0.96 | 18 | rs11191157 | 0.60 | A | −0.023 | 0.22 | 2.34×10⁻⁸ | *C10orf76* | Intron |
| 17 | 42,899,988 | 43,438,117 | 1 | 10 | 0.95 | 10 | rs12936234 | 0.20 | C | 0.020 | 0.47 | 4.28×10⁻⁹ | *NMT1* | Intron |

Abbreviations: A1 Freq, frequency of A1; A1, effect allele; Chr, chromosome; *N*, number; NA, not available. Gene symbols are italicized. [a]Number of genome-wide significant variants with *P* value of a two-sided test <5×10⁻⁸. Base pair position is based on the human genome assembly GRCh37 (hg19). 'Credible set ID': the ID of credible sets used to identify different credible sets in the same region. 'Gene': the genes affected by the variant using the Variant Effect Predictor tool. 'Annotation': the consequence of variants on the protein sequence using the Variant Effect Predictor tool.

Finally, we conducted pathway enrichment analysis using the Gene Set Analysis-Single-Nucleotide-Polymorphism-2 (GSA-SNP2)[28] to explore potential biological pathways associated with EduYears. Based on the GWAS summary statistics from EAS and EUR populations, we aimed to identify pathways significantly associated with EduYears in each population and subsequently compare the results to determine shared or distinct pathways between two populations. In total, 16 and 27 pathways were identified as significantly associated with EduYears in EAS and EUR populations, respectively (Fig. 2c). Among these significantly enriched pathways, 14 pathways were common across both populations, while 2 and 13 pathways exhibited significant enrichment exclusively in EAS and EUR populations, respectively.

**Cross-ancestry GWAS meta-analysis for EduYears**
To maximize the power of gene discovery, we conducted a cross-ancestry meta-analysis with EduYears GWAS summary statistics in EAS and EUR using METAL[29]. We obtained the publicly available summary statistics of EUR, including 766,345 samples and 10,101,242 variants from a previous large-scale GWAS performed by Lee et al.[14]. In total, 942,745 samples and 12,232,310 variants were included in the meta-analysis. We identified 315 lead SNPs at 102 genome-wide significant loci associated with EduYears (Fig. 1b and Supplementary Fig. 1d), all of which have been previously reported in EUR ancestry GWASs (313 variants in ref. 14 and rs9257925 and rs7224296 in ref. 10).

Additionally, we performed multi-ancestry meta-analysis (MAMA)[30], a GWAS meta-analysis method, which models differences in effect sizes, allele frequencies and LD patterns between populations and provides population-specific meta-analysis results. Using MAMA, we identified 94 independent genome-wide significant SNPs with EAS-specific meta-analysis (Fig. 1c and Supplementary Table 13), 2 of which were previously unreported for EduYears (rs2881903 and rs16930687); they were located beyond ±500 kb of the lead SNPs reported in previous EduYears GWAS (Supplementary Figs. 10 and 11)[10,14]. The MAMA EUR-specific meta-analysis found 357 independent genome-wide significant SNPs; however, all of them had been previously reported to be associated with EduYears[10] (Supplementary Fig. 12 and Supplementary Table 14).

To determine whether the genetic effects for EduYears were similar in two different populations (EAS and EUR), we estimated cross-ancestral genetic correlation using S-LDXR[31]. The genetic correlation across the EAS and EUR populations was 0.873 (s.e. 0.042).

**Assessment of transferability between EAS and EUR**
We investigated the transferability of EduYears genomic loci identified in the EUR population to the EAS population with the power-adjusted

transferability (PAT) ratio[32]. To consider differences in LD patterns, we first generated credible sets for the 246 genetic loci associated with EduYears from Lee et al.[14] study (*n* = 766,345). Based on the credible sets, the PAT ratio for EduYears for EUR to EAS was 0.62 (number of observed transferable loci divided by number of expected transferable loci in the EAS population = 95/153). This result indicates a relatively high transferability of GWAS loci associated with EduYears between EAS and EUR populations.

**Cross- and within-population fine-mapping for EduYears**
To further refine the seven genetic loci identified in the EAS GWAS meta-analysis, we performed within-population and cross-population fine-mapping in EAS and EUR populations using SuSiEx[33] with the 1000 Genomes (1KG) Project phase 3 samples as the LD reference panel. From the seven associated loci, we identified 8 credible sets in the EAS GWAS fine-mapping and 13 credible sets in the cross-population GWAS fine-mapping, with each credible set representing an independent association signal (Fig. 3, Tables 1 and 2, Supplementary Figs. 13–18 and Supplementary Tables 15–17). The potential causal variants often showed a higher posterior inclusion probability (PIP) in cross-population fine-mapping than in EAS population fine-mapping. For example, we fine-mapped one credible set for the locus on chromosome 17 from the GWAS meta-analysis of EAS (Fig. 3a), in which the variant with the maximum PIP was rs12936234 (PIP 0.20; gene *NMT1*) (Fig. 3c). We observed two different credible sets from the GWAS meta-analysis for EduYears in EUR population: rs2867316 (PIP 0.87; intergenic) and rs11871429 (PIP 0.51; gene *HIGD1B*) (Fig. 3b,d). In the cross-population fine-mapping, we identified three credible sets in which the variants with the maximum PIP were rs2867316 (PIP 0.90; gene *MAP3K14*), rs12948326 (PIP 0.71; gene *NMT1* and *PLCD3*) and rs11871429 (PIP 0.51; gene *HIGD1B*) (Fig. 3e). The maximum PIP in the credible set on chromosome 17 near 43.18 Mb was considerably larger in cross-population fine-mapping than in EAS population fine-mapping. The SNP rs11871429 was reported as a lead SNP in GWAS meta-analysis for EduYears in EUR population, and these three SNPs were located within ±500 kb of the 1,271 lead SNPs of GWAS meta-analysis for EduYears in EUR population[14].

We note that using external reference panels, whose LD patterns do not perfectly match LD in the discovery GWAS samples, may bias the results for fine-mapping. We therefore assessed the LD consistency of each locus between discovery and reference samples in EAS population using diagnostic tools provided by SuSiE-RSS[34]. The *s* values from SuSiE-RSS ranged from 0.007 to 0.027 across fine-mapped loci, and the diagnostic plots demonstrated high consistency of LD between the summary statistics and reference panel (Supplementary Fig. 19a–g).

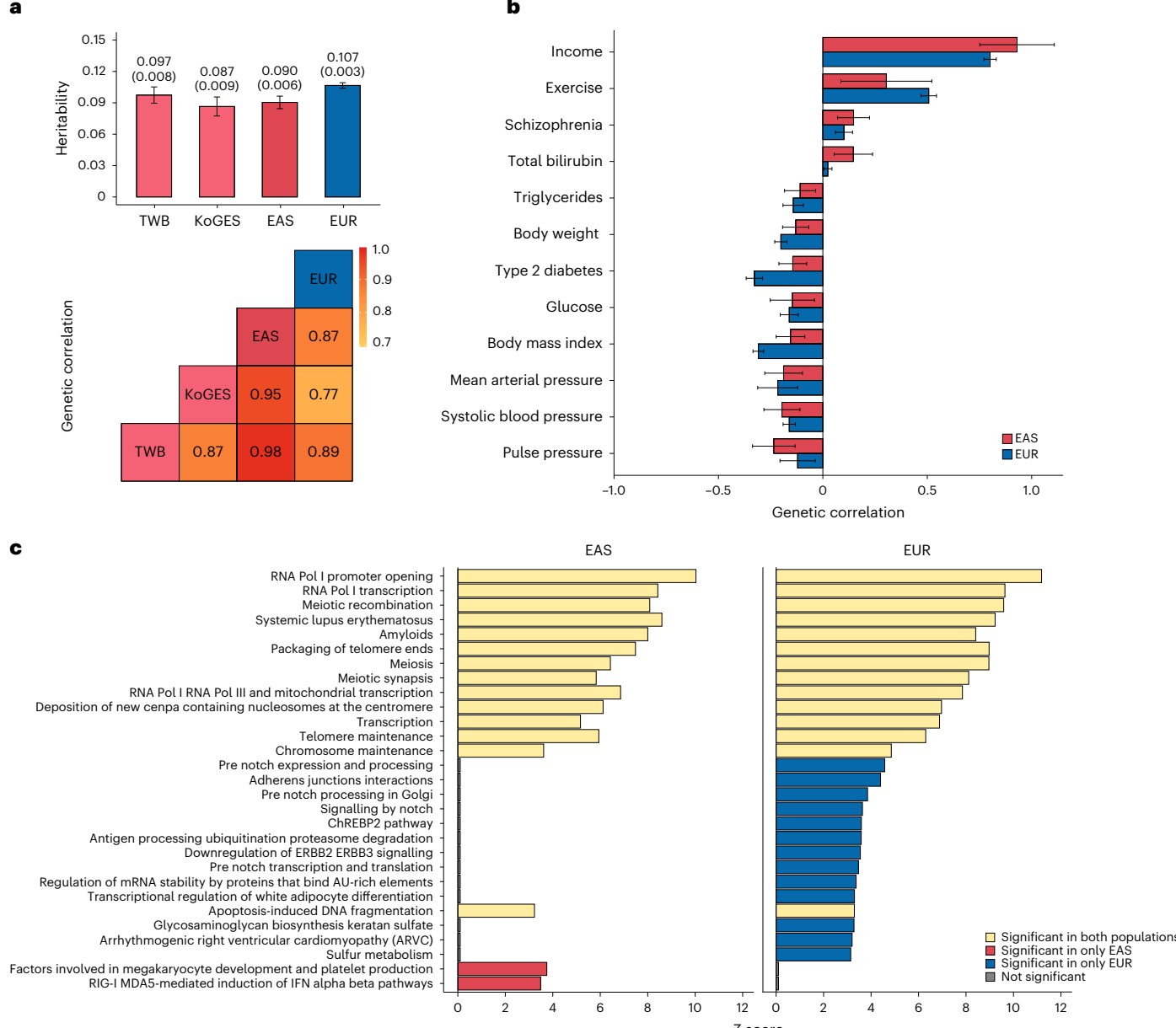

**Fig. 2 | Comparison of genetic architecture for EduYears between EAS and EUR. a**, SNP-based heritability and genetic correlation for EduYears in EAS and EUR. Top: we performed the LDSC to estimate the SNP-based heritability of EduYears in TWB ($N$ = 93,570), KoGES ($N$ = 71,662), EAS populations ($N$ = 165,232) and EUR populations ($N$ = 766,345). The $x$ axis represents the population, and the $y$ axis represents the SNP-based heritability. Bars indicate the estimates of SNP-based heritability for each population. Error bars (black line) indicate the 95% confidence intervals of the estimated SNP-based heritability. Bottom: we also performed the LDSC to estimate the genetic correlations between TWB, KoGES, EAS populations and EUR populations. The pairwise genetic correlations between TWB, KoGES, EAS populations and EUR populations are shown in red to yellow gradient. That is, colour close to red indicates a higher correlation, and colour close to yellow indicates a lower correlation. **b**, SNP-based genetic correlation between EduYears and other phenotypes in EAS and EUR. We showed 12 of 82 phenotypes with significant genetic correlation with EduYears (FDR <5%) in the EAS population. The $x$ axis is the genetic correlation between EduYears and other traits. Bars indicate the estimates of genetic correlation between EduYears and each trait. Error bars (black line) indicate the 95% confidence intervals of the estimated genetic correlation. All results including the sample size of each trait are presented in Supplementary Table 18. **c**, Pathway enrichment for EduYears in EAS and EUR. We showed significantly enriched pathways with a $q$ value <0.05 for EAS and EUR populations. The $x$ axis represents the $Z$-score, and the $y$ axis represents each individual pathway.

## Genetic correlation with other traits

To explore the genetic relationship between EduYears and other socioeconomic and health-related traits, we used LDSC[18] to estimate the genetic correlation between EduYears and 82 phenotypes for which GWAS summary statistics are available for the EAS population. We examined the genetic relationships between EduYears and the 82 phenotypes within each population and then checked the consistency across populations. We identified 12 phenotypes with significant pairwise genetic correlations with EduYears (FDR <5%) in the EAS population (Fig. 2b and Supplementary Table 18). In EAS, income showed the strongest positive genetic correlation with EduYears ($r_g$ = 0.93, $P$ = 9.87 × 10$^{-25}$) and pulse pressure showed the strongest negative genetic correlation with EduYears ($r_g$ = −0.24, $P$ = 6.20 × 10$^{-6}$). We obtained GWAS summary statistics from EUR samples for 64 phenotypes. By applying LDSC, EduYears showed the strongest positive genetic correlation with income ($r_g$ = 0.80, $P$ = 2.33 × 10$^{-700}$) and the

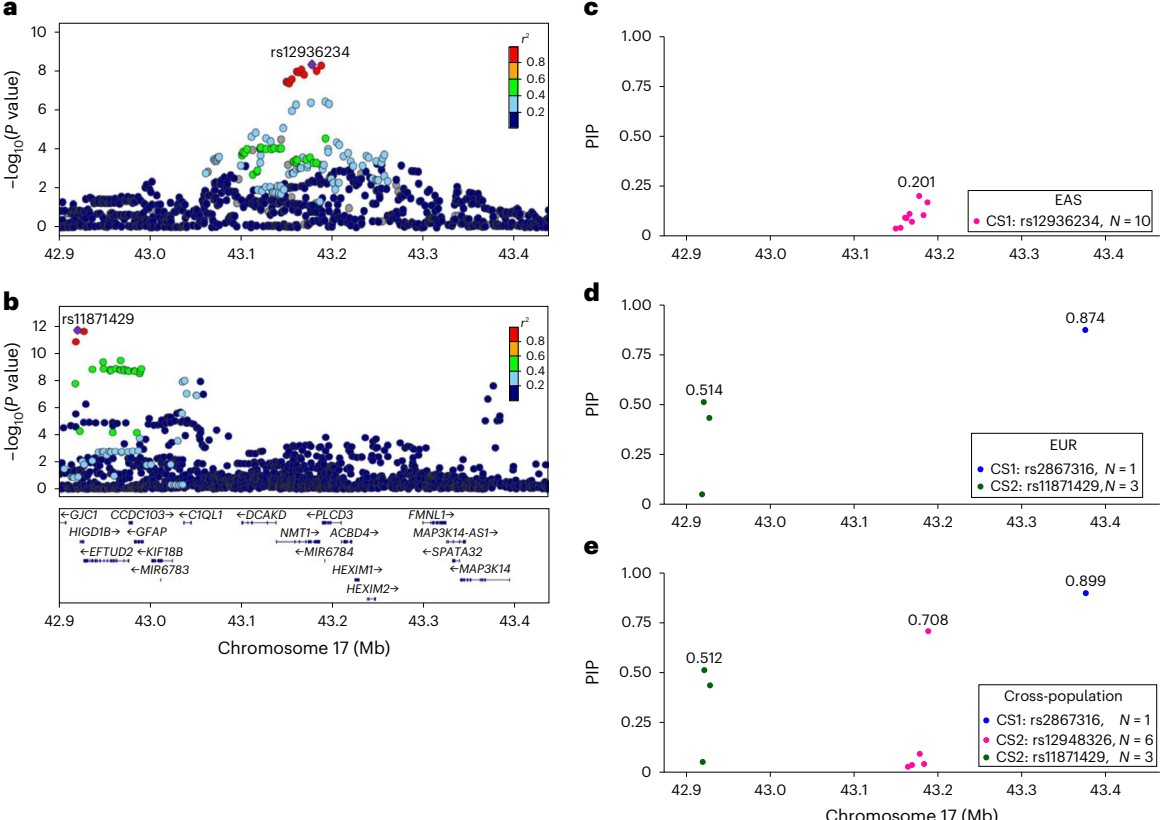

**Fig. 3 | Cross- and within-population fine-mapping for EduYears in a genome-wide significant locus (rs12936234) in EAS with EduYears GWAS in EUR by using SuSiEx. a**, Regional association plot in EAS. **b**, Regional association plot in EUR. **c**, Fine-mapping in EAS. **d**, Fine-mapping in EUR. **e**, Fine-mapping in cross-population. The *x* axis represents the chromosomal position for each panel. The *y* axis represents the $-\log_{10}(P$ value) for the association of variants with EduYears in EAS and EUR for **a** and **b**, respectively. For the meta-analysis of EduYears in EAS, an IVW fixed-effect model was used. All *P* values were calculated by a two-sided test and not corrected for multiple testing. In **a** and **b**,

LD estimates of surrounding SNPs with the index SNP are indicated by colour. The *y* axis represents the PIP of the SNPs included in the credible sets identified from fine-mapping in **c** to **e**. PIP is the probability of being the candidate causal variant for each SNP. We filtered out SNPs with $P > 5 \times 10^{-8}$. We marked the value of maximum PIP in each credible set, and we used different colours to distinguish each credible set. For example, there are one (pink points), two (green points and blue points) and three (green points, pink points and blue points) credible sets identified from EAS populations, EUR populations and cross-population, respectively.

---

strongest negative genetic correlation with type 2 diabetes ($r_g = -0.33$, $P = 4.24 \times 10^{-59}$) in the EUR population. We observed directional consistency of genetic correlations for most of these phenotypes in both populations, further confirming a similar genetic architecture of EduYears between the EAS and EUR populations.

## Polygenic prediction

To assess the predictive ability of polygenic scores (PGSs) for EduYears, we used our EAS summary statistics and the EUR summary statistics by Lee et al.[14] to construct PGSs and tested their predictive performance in three independent testing cohorts of EAS ancestry, including the Epidemiology of Mild Cognitive Impairment study in Taiwan (EMCIT), a Korean-based cohort, and the Chinese sample in the UK Biobank (UKBB), and one testing cohort of EUR ancestry, which is the National Institute on Aging Genetics Initiative for Late-Onset Alzheimer's Disease (NIA-LOAD). After stringent QC, we included 395, 2,622, 1,747 and 1,241 samples from the EMCIT, Korean-based cohort, UKBB and NIA-LOAD, respectively (Supplementary Table 1). We calculated the PGS for EduYears using two Bayesian polygenic prediction methods: PRS-CS[35] and PRS-CSx[36]. PRS-CS was individually applied to EAS and EUR EduYears GWAS to derive a single-population PGS. Meanwhile, PRS-CSx was integrated both EAS and EUR EduYears GWAS to generate a cross-population PGS. The cross-population PGS from PRS-CSx explaining up to 4.0% of the phenotypic variance in EduYears in the EAS cohorts. However, the cross-population PGS explained 6.1% of the

phenotypic variance in EduYears, equivalent to ancestry-matched PGS in the EUR cohort, possibly because it utilized a smaller GWAS of an unmatched population (Fig. 4 and Supplementary Table 19). Overall, the cross-population PGS from PRS-CSx showed better performance than the single-population PGS; the largest improvement was observed when applying PRS-CSx to the EAS testing cohorts. To investigate whether the improvement in predictive performance in the cross-population PGS was solely attributed to an increase in sample size or also influenced by ancestral diversity, we conducted an additional analysis by equating the sample sizes of EAS and EUR populations. Consistent with previous results, the cross-population PGS explained a greater proportion of phenotypic variance in EduYears than the EUR-derived PGS in the EAS cohorts (Supplementary Fig. 20 and Supplementary Table 20).

## Discussion

We present the largest-so-far EduYears GWAS in the EAS population and cross-ancestry GWAS meta-analysis across EAS and EUR populations for EduYears, including 176,400 samples of EAS genetic ancestry from TWB and KoGES and 766,345 samples of EUR genetic ancestry from previous studies, which enabled us to examine and compare the genetic architecture of EduYears across populations. Although the previous GWAS for EduYears had already reached a sample size of approximately 3 million, it was solely based on samples of EUR ancestry. Genetic studies of complex traits, including EduYears, have mostly been conducted in EUR population. This disparity is problematic because genomic

**Table 2 | Genome-wide significant loci for EduYears in cross-ancestry meta-analysis combining EAS and EUR populations**

| Chr | Start | End | Credible set ID | Credible set size | Total PIP | N of significant variants[a,*] | SNP with maximum PIP | Maximum PIP | A1 | Effect size (EAS) | A1 Freq (EAS) | Marginal P value (EAS) | Effect size (EUR) | A1 Freq (EUR) | Marginal P value (EUR) | Gene | Annotation |
|---|---|---|---|---|---|---|---|---|---|---|---|---|---|---|---|---|---|
| 2 | 161,721,597 | 162,351,261 | 1 | 2 | 0.99 | 2 | rs11678980 | 0.93 | A | −0.020 | 0.39 | $4.19 \times 10^{-9}$ | −0.017 | 0.43 | $3.69 \times 10^{-24}$ | NA | NA |
| 4 | 180,613,679 | 181,136,169 | 1 | 4 | 0.99 | 4 | rs4861427 | 0.38 | C | 0.020 | 0.47 | $6.81 \times 10^{-9}$ | 0.005 | 0.48 | $3.66 \times 10^{-3}$ | NA | NA |
| 5 | 93,838,858 | 94,392,030 | 1 | 15 | 0.95 | 14 | rs255338 | 0.18 | C | 0.022 | 0.73 | $1.11 \times 10^{-8}$ | 0.007 | 0.90 | $6.08 \times 10^{-3}$ | MCTP1 | Intron |
| 5 | 106,947,725 | 107,455,182 | 1 | 2 | 0.95 | 2 | rs7708343 | 0.51 | A | −0.021 | 0.32 | $4.46 \times 10^{-9}$ | −0.012 | 0.11 | $2.28 \times 10^{-6}$ | FBXL17 | Intron |
| 5 | 106,947,725 | 107,455,182 | 2 | 15 | 0.96 | 2 | rs6868799 | 0.17 | A | 0.009 | 0.36 | $9.40 \times 10^{-3}$ | 0.009 | 0.35 | $3.13 \times 10^{-8}$ | FBXL17 | Intron |
| 6 | 27,515,505 | 29,611,229 | 1 | 3 | 0.96 | 3 | rs9461540 | 0.39 | G | −0.027 | 0.17 | $6.19 \times 10^{-10}$ | NA | NA | NA | GABBR1 | Upstream gene |
| 6 | 27,515,505 | 29,611,229 | 2 | 35 | 0.96 | 25 | rs112644424 | 0.13 | T | −0.022 | 0.28 | $2.33 \times 10^{-9}$ | −0.014 | 0.08 | $2.45 \times 10^{-6}$ | NA | NA |
| 10 | 103,385,878 | 104,057,295 | 1 | 26 | 0.96 | 26 | rs3758551 | 0.07 | G | −0.013 | 0.77 | $1.06 \times 10^{-3}$ | −0.012 | 0.60 | $6.24 \times 10^{-14}$ | GBF1, PITX3 | Upstream gene |
| 10 | 103,385,878 | 104,057,295 | 2 | 4 | 0.98 | 4 | rs11191193 | 0.70 | G | −0.021 | 0.23 | $6.89 \times 10^{-8}$ | −0.017 | 0.34 | $5.89 \times 10^{-23}$ | C10orf76 | Intron |
| 10 | 103,385,878 | 104,057,295 | 3 | 3 | 0.98 | 0 | rs11598489 | 0.54 | T | NA | NA | NA | 0.023 | 0.04 | $2.65 \times 10^{-7}$ | NA | NA |
| 17 | 42,899,988 | 43,438,117 | 1 | 2 | 0.97 | 1 | rs2867316 | 0.90 | T | 0.004 | 0.62 | $2.84 \times 10^{-1}$ | −0.010 | 0.32 | $2.32 \times 10^{-8}$ | MAP3K14 | Intron |
| 17 | 42,899,988 | 43,438,117 | 2 | 7 | 0.97 | 6 | rs12948326 | 0.71 | G | −0.020 | 0.53 | $4.90 \times 10^{-9}$ | −0.004 | 0.39 | $1.04 \times 10^{-2}$ | NMT1, PLCD3 | Downstream gene |
| 17 | 42,899,988 | 43,438,117 | 3 | 3 | 1.00 | 3 | rs11871429 | 0.51 | G | −0.005 | 0.33 | $1.45 \times 10^{-1}$ | −0.014 | 0.21 | $1.73 \times 10^{-12}$ | HIGD1B | Upstream gene |

Abbreviations: A1 Freq, frequency of A1; A1, effect allele; Chr, chromosome; N, number; NA, not available. Gene symbols are italicized. [a]Number of genome-wide significant variants with P value of a two-sided test <5×10⁻⁸. *at least in one population. Base pair position is based on the human genome assembly GRCh37 (hg19). 'Credible set ID': the ID of credible sets used to identify different credible sets in the same region. 'Gene': the genes affected by the variant using the Variant Effect Predictor tool. 'Annotation': the consequence of variants on the protein sequence using the Variant Effect Predictor tool.

discoveries may not be transferable across populations, even though it is assumed that they share the underlying biological mechanisms[37], and our study helps to fill this gap.

This study provides several important findings regarding the genetics of EduYears. First, we observed high positive genetic correlations of EduYears within the EAS population ($r_g = 0.87$) and between the EAS and EUR populations ($r_g = 0.87$). This suggests a comparable degree of shared genetic component for EduYears within the EAS and between EAS and EUR. To benchmark the EAS–EUR cross-population $r_g$ for EduYears against other traits, we extracted EAS–EUR cross-population $r_g$ for 31 other traits from Shi et al.[31] as a reference. Remarkably, the cross-population $r_g$ for EduYears closely aligns with the median of EAS–EUR cross-population $r_g$ across the 31 traits (median $r_g = 0.88$; range 0.342–1.05). While the EAS–EUR cross-population $r_g$ for EduYears is lower than that for schizophrenia (EAS–EUR cross-population $r_g = 0.945$), it is considerably higher than major depressive disorder (EAS–EUR cross-population $r_g = 0.342$) and comparable to other physiological traits (EAS–EUR cross-population $r_g = 0.897$ for height) and molecular phenotypes (EAS–EUR cross-population $r_g = 0.875$ for haemoglobin A1c). The SNP-based heritability was similar within the EAS population ($9.7 \pm 0.8\%$) for TWB and ($8.7 \pm 0.9\%$) for KoGES and between EAS ($9.0 \pm 0.6\%$) and EUR ($10.7 \pm 0.3\%$) populations[14]. The larger s.e. of heritability estimates in the EAS reflected the smaller EAS GWAS sample sizes relative to the EUR GWAS. The direction of genetic effects showed consistency between TWB and KoGES for genome-wide significant loci identified in both studies, and there was no substantial difference in the allele frequency between TWB and KoGES, except for one locus (near *ALDH2*) on chromosome 12. We have confirmed that the observed heterogeneity in the *ALDH2* region may be linked to possible shared genetic component and gene-environment interaction between alcohol drinking and EduYears, in the Korean population. This finding suggests that studying diverse populations can bring insights in identifying gene–environment associations. To facilitate cross-population comparisons, we investigated the transferability of EduYears loci between EAS and EUR populations using the PAT ratio approach[32], which considers the potential limitation of statistical power in the EAS population compared with EUR. Our findings indicate a relatively high transferability of EduYears loci identified in the EUR population to the EAS population.

Indeed, consistent with the high genetic correlation and transferability observed between EAS and EUR populations, our partitioned heritability and LDSC-SEG analyses[24,25] showed similar results for both populations. Additionally, the pathway enrichment analysis demonstrated shared biological pathways between the EAS and EUR populations. We showed that 14 pathways were significantly associated with EduYears in both populations. These findings suggest the consistent involvement of specific biological pathways in the genetic basis of educational attainment, regardless of ancestry. Furthermore, these shared pathways underscore their potential importance in contributing to the association with educational attainment across diverse populations.

Our cross-ancestry meta-analysis identified genome-wide significant loci associated with EduYears that were not previously reported. We found 102 genome-wide significant loci for EduYears by cross-population meta-analysis using METAL[29] and 94 independent SNPs in an EAS-specific cross-population meta-analysis using MAMA[30], 2 of which were not reported previously (rs2881903 and rs16930687). We check the MAF values for these two SNPs in EAS and EUR populations. The MAF values of rs2881903 and rs16930687 were 5.2% versus 8.8% and 2.3% versus 0.7% in EAS and EUR[14], respectively. The nearest gene for rs2881903 was *FAM81B*, and the nearest gene for rs16930687 was a processed pseudogene (*PR11-224P11.1*). Both genes have not been reported to be associated with other traits in GWAS. Further studies of the function of these genes will help elucidate their biological mechanisms for EduYears.

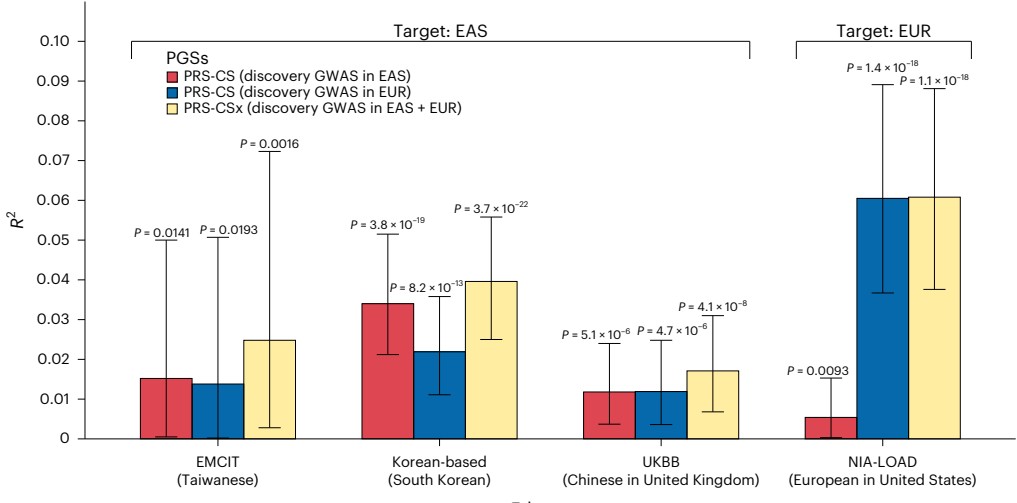

**Fig. 4 | Polygenic prediction of EduYears in the EMCIT ($n$ = 395; EAS), Korean-based cohort ($n$ = 2,622; EAS), Chinese samples in the UKBB ($n$ = 1,747; EAS), and the NIA-LOAD ($n$ = 1,241; EUR).** The $x$ axis shows the testing cohort, and the $y$ axis is the partial $R^2$ for the PGS. Bars indicate the partial $R^2$ for the PGS of each cohort. Error bars (black line) indicate the 95% confidence intervals of the partial $R^2$ for PGS. PGSs were derived from the GWAS meta-analysis for EduYears in EAS by using PRS-CS (discovery GWAS in EAS), the GWAS meta-analysis for EduYears in EUR by using PRS-CS (discovery GWAS in EUR), and both GWAS meta-analysis for EduYears, including EAS and EUR, by using PRS-CSx (discovery GWAS in EAS and discovery GWAS in EUR). We adjusted for the birth year, sex, birth year by sex interaction, and top ten PCs in all models. The two-sided $P$ value of the partial $R^2$ was derived from a likelihood ratio test comparing the goodness of fit of the models with and without PGS, which were annotated above the error bars.

Previous studies have found that population diversity may improve fine-mapping resolution[38,39] by capitalizing on increased GWAS sample sizes and LD differences across ancestries. Our study provided compelling evidence to support this hypothesis by demonstrating that cross-population fine-mapping substantially increased the maximum PIP compared with within-population fine-mapping. An example from this study is the genome-wide significant locus near 43.18 Mb on chromosome 17. While an association signal was not found in the EUR population, the maximum PIP in the credible sets increased from 0.201 in EAS population fine-mapping to 0.708 in cross-population fine-mapping.

Okbay et al. investigated the genetic correlation of EduYears with 14 traits and found positive genetic correlations between EduYears and intracranial volume, bipolar disorder, schizophrenia, cognitive performance, and height and negative genetic correlations between EduYears and Alzheimer's disease, neuroticism and body mass index in the EUR population[9]. In this study, we expanded the investigation to a total of 82 socioeconomic and health-related traits for which GWAS summary statistics were available for the EAS population. The effect directions of genetic correlations between EduYears and socioeconomic and health-related traits were consistent between EUR and EAS populations. EduYears showed significant genetic correlations with 12 out of the 82 traits in the EAS population, all of which were also significant in the EUR population with concordant directions. This result further supports a similar genetic architecture for EduYears between the two ancestries and highlights the shared genetic components between EduYears and various socioeconomic and health-related outcomes.

Finally, we evaluated the predictive performance of ancestry-specific and cross-population PGSs in EAS and EUR cohorts independent of the discovery GWAS. As expected, the EAS-specific PGS showed better prediction power than the EUR-specific PGS for three independent EAS testing cohorts. The performance of PGS for within population prediction based on EAS GWAS is comparable to that based on EUR GWAS ($n$ = 117,922) with a similar sample size (EAS EduYears PGS $R^2$ = 1.5–3.4% in the EMCIT and the Korean-based cohort (EAS target sample) versus EUR EduYears PGS $R^2$ = 2.6–2.8% in the National Longitudinal Study of Adolescent to Adult Health and the Health and Retirement Study (EUR target sample)[10,40]). In the UKBB Chinese samples, which are genetically EAS but may be environmentally close to EUR, both the EUR-based PGS and EAS-based PGS showed lower predictive power than in other EAS cohorts in our study or previous studies[10,14]. The best predictive performance in the EAS testing cohorts was achieved by the cross-population PGS derived from PRS-CSx. This result replicated previous studies showing that multi-ancestry PGS demonstrated an improved prediction performance relative to ancestry-matched PGSs[41,42]. However, the NIA-LOAD cohort, comprising EUR individuals, showed equivalent explanatory power between EUR-specific PGS and cross-population PGS because the gain from leveraging a smaller GWAS in an unmatched population may be limited. Even with the same sample size, the cross-population PGS consistently outperformed the EUR-derived PGS in the EAS testing cohorts. This observation suggests that population diversity enhanced predictive performance. Through PGS analyses, we explored the transferability of PGS between EAS and EUR populations, which is critical information regarding the utility of PGS. Furthermore, our PGS analyses also indicated the advantages of ancestral diversity over a single population in PGS construction[36].

This study had several limitations. One limitation is that the education level as measured in our study may not be equal to the actual education years. Instead of collecting the number of years of education, EduYears were usually collected using a self-reported questionnaire of educational attainment. Regardless of whether the participants graduated from a specific educational level, they might be classified into the same category. For example, the elementary school category in TWB indicates whether the participants had attended or graduated from elementary school, but the actual number of years of education they received may range from 0 to 6 years. On the other hand, the difference in the years of compulsory education, described in detail in Supplementary Note, required in Taiwan and South Korea might limit the phenotypic variation in EduYears. Another limitation might be the relatively small EAS sample size compared with GWAS in the EUR population. To mitigate this limitation, we conducted a meta-analysis of two cohorts of EAS ancestry to increase the power of the EduYears GWAS for EAS. Indeed, we obtained more genomic signals from GWAS meta-analysis in EAS than GWAS in either TWB or KoGES. However, compared with the largest GWAS for EduYears in EUR, considerably

fewer genomic loci in EAS were identified (7 loci in EAS versus 3,952 loci in EUR) and all 7 loci reported in EAS were previously reported in the EUR GWAS. The absence of previously unreported loci in the EduYears GWAS in EAS compared with previous EUR GWAS reflects the lower power for gene discovery with the current sample size in TWB and KoGES. However, we expect to obtain more insight into the genetic basis of EduYears in the EAS population as the sample size increases with more samples from TWB and KoGES, as well as the inclusion of more EAS cohorts.

As previous studies and our results suggest, EduYears shows phenotypic correlations and shares genetic components with multiple traits and diseases relevant to medical research, including cognitive function, neurodegeneration and psychiatric disorders[43–46], and findings on genetic overlaps between EduYears and health outcomes may shed light on the genetic basis of these relevant health outcomes. However, the link between EduYears and these health outcomes varies with context (such as nationality)[47,48] and the impact of EduYears on health outcomes is probably via complex mechanisms like mediation and interaction between genetic and environmental factors[47]. To this point, we would like to highlight that while understanding the genetic basis of EduYears (as a proxy phenotype) may improve our insights of other relevant health outcomes, our results do not support any immediate medical or clinical applications, such as polygenic prediction in direct-to-consumer services[49,50].

In conclusion, our study, as the first large-scale educational attainment GWAS in EAS, provides insights into the genetic architecture and biological mechanisms of EduYears across EAS and EUR populations through gene discovery, SNP-based heritability and genetic correlation analysis, functional analysis and pathway enrichment analysis. Furthermore, we demonstrated that cross-population GWAS improved fine-mapping resolution and PGS prediction performance in the context of educational attainment. These results underscore the importance of combining diverse population cohorts in genetic studies. As the largest previous GWAS of EduYears was limited to the EUR population, our EduYears GWAS in EAS and cross-population GWAS meta-analysis enhance our comprehension of the genetic basis of EduYears and facilitate the transfer of genetic insights for EduYears across populations.

## Methods

### Study selection
This study has been approved by the ethics committee of National Health Research Institutes, Taiwan (TWB; EC1090402-E and EC1110608-E) and Seoul National University Bundang Hospital, South Korea (KoGES; X-2107-699-902).

**TWB.** TWB is a population-based prospective cohorts study, which was planned to recruit 200,000 volunteers between 20 and 70 years of age with no prior diagnosis of cancer from 29 recruitment centres across Taiwan[16] (see URLs). In total, TWB has recruited 159,195 participants since 2012. Baseline characteristic data were collected from structured interviews, physical measurements, biomarkers and genetic data. We obtained genome-wide genotype data from two customized chips, including 27,719 samples in the TWB v1 custom array and 81,236 samples in the TWB v2 custom array. The TWB v1 custom array (batch 1) was designed on the basis of the Thermo Fisher Axiom Genome-Wide CHB Array with customized contents in 2011, and the TWB v2 custom array (batch 2) was designed by Thermo Fisher Scientific in 2017 on the basis of whole-genome sequencing data from 946 TWB samples with customized contents[51].

**KoGES.** The KoGES is a large prospective cohort study initiated by the National Institute of Health, South Korea. KoGES provides epidemiological and genetic data from three population-based cohorts: Ansan/Ansung, Health Examinee and the Cardiovascular Disease Association

study. Ansan/Ansung is a community-based cohort that recruited 10,030 individuals aged 40–69 years living in Ansan or Ansung. Health Examinee is an urban-based cohort study that recruited 173,208 individuals aged 40–79 years between 2004 and 2013 at a hospital health check-up centre. The Cardiovascular Disease Association study is a rural-based cohort conducted between 2005 and 2011 and recruited 28,337 individuals aged 40–69 years. From these three cohorts, we obtained genome-wide genotype data from the Korea Biobank Array, which is a customized Korean-specific chip[52,53]. In total, 71,678 individuals with genotypic and phenotypic information were included in this study.

### Genotype data QC and imputation
We conducted stringent pre-imputation QC, followed by the PBK genotype QC project pipeline (see URLs), for samples in TWB batch 1, TWB batch 2 and KoGES. First, we included samples with a call rate >0.98 and variants with a call rate >0.98. We then filtered out variants that were duplicated, monogenic or incorrectly mapped to a genomic position. Using a random forest model with the top six principal components (PCs) and the 1KG Project phase 3 data as ref. 54, we classified genetic ancestry and identified samples with a predicted probability of EAS ancestry >0.8. When we estimated the PCs with LD pruning at $r^2 = 0.1$, we removed multi-allelic and strand ambiguous SNPs, SNPs with call rate <0.98, SNPs with MAF >5%, and SNPs located in long-range LD regions (chromosome 6: 25–35 Mb; chromosome 8: 7–13 Mb). We then excluded samples with mismatched genetic and self-reported sex, as well as samples with heterozygosity rates outside six standard deviations from the sample average. We also excluded population outliers by conducting in-sample PC analysis in three rounds. We excluded samples with any of the top ten PCs that were more than six standard deviations away from the sample average in each round of the in-sample PC analysis. Finally, we included homogeneous EAS samples and discarded variants with call rate <0.98 and Hardy–Weinberg equilibrium $P$-value <$10^{-10}$. After pre-imputation QC, we performed imputation independently for TWB batch 1, TWB batch 2 and KoGES using Eagle v2.4 (ref. 55) for pre-phasing and Minimac4 for genotype imputation, with the 1KG Project phase 3 EAS data as the reference panel[56].

### Phenotype: EduYears
The education system in Taiwan is similar to that in South Korea (for cohort details, see Supplementary Note). EduYears was collected from different questionnaires using a multiple-choice question in the TWB and KoGES when the participant was 30 years old or older. To ensure comparability across cohorts, including TWB, KoGES and cohorts in GWAS meta-analysis in EUR population, we mapped each category in these questions to the International Standard Classification of Education (ISCED) category. We then imputed each ISCED category to the number of years of schooling, which is referred to as EduYears. We have summarized the questions and results of mapping from the ISCED Level to EduYears in Supplementary Table 21.

### Genetic association analysis
We performed genetic association analyses on EduYears using post-QC imputed genotype data and a two-step whole-genome regression model implemented in Regenie v2.2.4 (ref. 57), which accounts for sample relatedness and population structure. We excluded duplicate samples by randomly removing one sample from each pair in the two-step whole-genome regression models for TWB and KoGES. We adjusted for the birth year (BY), $BY^2$, $BY^3$, sex, BY by sex interaction, $BY^2$ by sex interaction, $BY^3$ by sex interaction, and the top ten PCs, based on previous GWAS for EduYears[14]. The top ten PCs were included as covariates to control for population stratification. We used the GWAS summary statistics derived from Regenie[57] as the main result and utilized them for all downstream analyses, except for the analyses using LDSC[18] and LDSC-based methods such as S-LDXR[31] and stratified LDSC.

For analyses using LDSC-based methods (for example, LDSC, S-LDXR, stratified LDSC and so on), we separately performed association analyses using linear regression implemented in PLINK v2.0 (ref. [58]) in unrelated samples of TWB and KoGES. We estimated genetic relatedness to check for family relationships using PLINK[58] with kinship coefficients of 0.0884 and 0.354 as thresholds for second-degree relations and duplicate samples, respectively. We randomly excluded one sample from each pair of second-degree or more closely related relatives within TWB batch 1, TWB batch 2 and KoGES independently. Across batches in TWB, we also excluded batch 2 samples from each pair of duplicated samples. We then performed association tests adjusted for BY, $BY^2$, $BY^3$, sex, BY by sex interaction, $BY^2$ by sex interaction, $BY^3$ by sex interaction, and the top ten PCs on the remaining unrelated individuals for TWB batch 1, TWB batch 2 and KoGES.

## GWAS meta-analyses in EAS population
Before performing the following meta-analyses, we first filtered the variants in individual biobank association summary statistics (TWB batch 1, TWB batch 2 and KoGES) by imputation INFO >0.6 and MAF >0.5% for Regenie whole-genome linear regression analyses and by imputation INFO >0.8 and MAF >1% for PLINK linear regression analyses. We first synthesized TWB batch 1 and TWB batch 2 GWASs using an inverse-variance-weighted (IVW) fixed-effect model implemented in METAL 2020-05-05 (ref. [29]). We conducted a GWAS meta-analysis of EduYears in EAS population, including TWB and KoGES, using an IVW fixed-effect model. In the meta-analyses of EduYears, we used the ID obtained from imputation[56] as unique identifiers for each variant, and we checked the heterogeneity in effect size using Cochran's Q test implemented in METAL[29]. We estimated Fst between TWB and KoGES to measure population differentiation due to genetic structure[59]. We removed variants with inconsistent allele on the same strand. We also removed variants that were not included in both biobanks. Subsequently, we annotated the reference SNP with the dbSNP build 155 data from the National Center for Biotechnology Information Search database[60] using SnpSift v4_3t_core[61].

## Heterogeneity of genetic effects within EAS population
To identify underlying factors contributing to heterogeneity between TWB and KoGES, we performed the following procedures.

**Phenome-wide association study lookup.** To investigate the pleiotropic effects of variants showing heterogeneity, we conducted a search in the KoGES PheWeb (see URLs).

**Global and local genetic correlation analyses.** To explore the relationship between alcohol-related traits and EduYears, we performed global and local genetic correlation analyses using KoGES data. The global genetic correlation was estimated using LDSC v1.0.1 (ref. [18]), while the local genetic correlation within specific genomic regions was assessed using LAVA[62]. The details are summarized in Supplementary Note.

**Stratified GWAS analysis.** In KoGES, individuals were classified as drinkers if they had a history of past or current alcohol consumption, and non-drinkers if they had no history of alcohol consumption. We then performed genetic association analyses for EduYears using Regenie v2.2.4 (ref. [57]), adjusting for BY, $BY^2$, $BY^3$, sex, BY by sex interaction, $BY^2$ by sex interaction, $BY^3$ by sex interaction, and the top ten PCs.

## Heritability and genetic correlation analyses
We performed LDSC v1.0.1 (ref. [18]) to estimate the SNP-based heritability of EduYears in the TWB, KoGES, EAS and EUR populations[14] using population-specific LD scores based on the 1KG Project phase 3 data. We applied LDSC[18] and S-LDXR v0.3-beta[31] to estimate the genetic correlations for within-EAS and cross-population genetic correlations

between EAS and EUR populations, respectively. We used the default LD scores for the EAS and EUR populations provided by S-LDXR[31] as reference panels to estimate the cross-ancestral genetic correlation. For comparability and unbiased estimation, we used GWAS summary statistics derived from linear regression models implemented in PLINK to perform LDSC[18] and S-LDXR[31], which excluded strand ambiguous SNPs and variants with imputation INFO <0.8 and MAF <1%.

## eQTL analysis
To investigate the influence on gene regulation of SNPs associated with EduYears, we performed an eQTL analysis using gene expression data implemented in FUMA v1.3.7 (ref. [22]) with brain tissue expression data from the GTEx v8 database. We then performed gene mapping for significant SNP-gene pairs with an FDR <5%.

## Gene-based and gene set enrichment analyses
Gene-based and gene set analyses were performed using MAGMA gene-property analyses[20,21] implemented in FUMA v1.3.7 (ref. [22]) to identify the genes and gene sets related to EduYears. Gene-based analysis was performed by mapping SNPs to 18,123 protein-coding genes using the SNP-wide mean model. Next, gene set analysis was conducted with 10,678 gene sets, including curated gene sets and GO terms from MsigDB v6.2. We employed a competitive test to determine whether genes in a gene set were more strongly associated with EduYears than the other genes. We then applied a Bonferroni correction to all tested genes and gene sets to account for multiple comparisons. MAGMA gene property analysis was performed using test statistics obtained from gene-based and gene set analyses.

## Partitioned heritability analysis
Based on GWAS summary statistics of EAS samples, we used LDSC-SEG v1.0.1 (ref. [25]) to prioritize tissues and cell types relevant to EduYears. We partitioned genome-wide SNP heritability into 97 baseline-LD annotations introduced by Gazal et al.[26] and 9 tissue-specific categories as specified by Finucane et al.[24]. We used LD scores for the EAS and EUR populations using the 1KG Project phase 3 data provided by LDSC GitHub repository as a reference (see URLs).

## Pathway enrichment analyses in EAS and EUR populations
We applied GSA-SNP2 (ref. [28]) based on all $P$ values from both EAS and EUR GWAS to detect biological pathways associated with EduYears. GSA-SNP2 employs the $Z$-statistics of the random set model, assessing pathways by combining adjusted gene scores for SNP counts in each gene using a monotone cubic spline trend curve. We evaluated gene set enrichment using the MSigDB C5 collection v5.2 database[63,64]. For the detailed options regarding the genes and pathways in the analysis, the race was selected as ancestry-matched (EUR or EAS), the reference genome version was set as GRCh37 (hg19), the padding size for genes was set to 20 kb, and the pathway size window was chosen as 10–200. Significantly enriched pathways were defined as those with $q$ value <0.05.

## Cross-ancestry GWAS meta-analyses in EAS and EUR populations
We obtained summary statistics for all variants that passed QC filters in the GWAS meta-analysis for EduYears in EUR population from all discovery cohorts except 23andMe[14], which included 766,345 participants of EUR ancestry with 10.1 million genetic variants. Next, we conducted a cross-ancestry GWAS meta-analysis to synthesize EUR and EAS data using an IVW fixed-effect model implemented in METAL[29], in which genomic control correction was applied to the EUR and EAS data. To evaluate the effect size heterogeneity between the two populations, we examined Cochran's $Q$ statistic implemented in METAL[29]. We applied MAMA[30] to account for potential differences between the EAS and EUR populations in effect size, allele frequency and LD.

We used the 1KG Project phase 3 data as a reference panel[54] to calculate the LD score for the EAS and EUR populations. Next, we filtered out variants with MAF <0.5%, and the remaining options for running the meta-analysis were set to default values. Okbay et al.[10] reported an updated meta-analysis of EduYears in a sample of 3,037,499 individuals in 2022, which was nearly three times larger than that reported by Lee et al. in 2018 ($n = 1,131,881$)[14]. However, there is an access limitation to publicly released data for both GWAS results. The sample size in the publicly released data from Lee et al. ($n = 766,345$) was slightly larger than that reported by Okbay et al. ($n = 765,283$); therefore, we used the GWAS results from Lee et al. in this study.

## Assessment of transferability

To assess the transferability of EduYears-associated loci between EAS and EUR populations, we employed the PAT ratio approach[32]. We initiated the analysis with 246 loci identified from publicly available EUR summary statistics ($n = 766,345$) by Lee et al.[14]. For each locus, we generated credible sets, incorporating lead SNPs and proxy SNPs, using the same criteria of the study by Huang et al.[32]. Specifically, we included SNPs within a 50-kb window of the lead SNP with $r^2 \geq 0.8$ and $P < 100 \times P_{lead}$ using the 1KG Project phase 3 EUR data as the reference panel. A locus was considered 'transferable' if at least one variant within its credible set exhibited an association with EduYears in EAS ($P < 0.05$) and demonstrated the same effect direction as observed in EUR. To estimate statistical power, we used the default parameter ($\alpha = 0.05$) and the summed-up power estimates for all published loci to obtain the expected number of transferable loci. Finally, by dividing the observed number of loci by the expected number of loci, we calculated the PAT ratio to estimate the transferability of EduYears loci between the EAS and EUR populations.

## Fine-mapping analysis

We used the SuSiEx approach[33], which builds on the Sum of Single Effects model, to perform within-EAS fine-mapping for EduYears in genome-wide significant loci and cross-population fine-mapping integrating EAS and EUR EduYears GWAS[14]. The GWAS summary statistics in EAS population were derived from the two-step whole-genome regression models implemented in Regenie[57], which filtered out variants with imputation INFO <0.6 and MAF <0.5%. The 1KG Project phase 3 data were used as the reference panel to calculate the LD matrix in the corresponding populations. We extended the region of a significant locus, identified through FUMA[22], by adding 250 kb to each side, if the region was less than 1 Mb. We then identified a 95% credible set in each region with the maximum number of the causal signals set to 10 and the default settings in the remaining options. We showed regional plots for these genome-wide significant loci in both EAS and EUR populations using LocusZoom v0.9.6 (ref. [65]). We also showed plots of PIPs in all credible sets identified from within- and cross-population fine-mapping, after filtering out variants with $P > 5 \times 10^{-8}$. Moreover, as the mismatch of the LD patterns between the reference panel and the GWAS discovery sample may bias the results for fine-mapping, we applied the SuSiE-RSS model[34] to evaluate the consistency of the LD using the susieR package v0.12.10 implemented in R v4.2.1 (ref. [66]). A larger 's' metric from SuSiE-RSS implies a strong inconsistency between GWAS summary statistics and the LD matrix from the reference panel. We also constructed diagnostic plots to compare the observed $z$-scores against the expected $z$-scores for SNPs included in the fine-mapping.

## Genetic correlation analysis with other traits

We estimated the cross-trait genetic correlation between EduYears and other traits by using LDSC v1.0.1 (refs. [18],[67]). We used publicly available GWAS summary statistics of socioeconomic and health-related traits for the EAS and EUR populations. A full list of GWAS summary statistics used in the analysis can be found in Supplementary Table 18.

For both populations, we downloaded LD scores calculated from the 1KG Project phase 3 data via the LDSC GitHub repository (see URLs). We then applied FDR correction to control for false positive discoveries.

## Polygenic prediction

We assessed the predictive ability of PGSs derived from the current EAS genome-wide meta-analysis for EduYears and the EUR GWAS for EduYears[14] by using three testing cohorts of EAS ancestry, which are the EMCIT, a Korean-based cohort and the Chinese samples in UKBB, and one testing cohort of EUR ancestry, which is the NIA-LOAD. These testing cohorts are summarized in Supplementary Note.

We constructed PGSs for EduYears using two Bayesian polygenic prediction methods, PRS-CS v1.0.0 (ref. [35]) and PRS-CSx v1.0.0 (ref. [36]). The advantages of PRS-CS are robustness to varying genetic architectures, accurate LD modelling and computational efficiency. The posterior SNP effect sizes in PRS-CS were inferred from the EAS and EUR GWAS meta-analysis for EduYears. PRS-CSx can be considered as an extension of PRS-CS, which improves cross-population polygenic prediction by integrating GWAS summary statistics from multiple ancestry groups. The posterior SNP effect sizes in PRS-CSx were inferred from both the EAS and EUR GWAS meta-analyses for EduYears[14]. We then synthesized the SNP effect sizes across populations using an IVW meta-analysis of population-specific posterior effect size estimates. The 1KG Project phase 3 samples (EAS ($n = 504$), EUR ($n = 503$)) that matched the ancestry of the discovery samples were used as external LD reference panels. We fixed the global shrinkage parameter to be 0.01 in both PRS-CS and PRS-CSx, which is suitable for highly polygenic traits.

We evaluated the prediction accuracy of PGS using the partial $R^2$ in each testing cohort, which was implemented in R v4.2.1. We adjusted for BY, sex, and BY by sex interaction, and the top ten PCs in all models. The $P$ value of the partial $R^2$ was derived from a likelihood ratio test comparing the goodness of fit of the models with and without PGS. To infer confidence intervals, we used the boot package[68] in R with 1,000 bootstrap replicates over samples.

The details for the analyses under the same sample sizes are summarized in Supplementary Note.

## Reporting summary

Further information on research design is available in the Nature Portfolio Reporting Summary linked to this article.

# Data availability

A detailed description of the availability and application process of the individual-level TWB data can be found at https://www.biobank.org. tw/english.php. Briefly, TWB made available the individual-level data and biological samples from the participants of the prospective cohort study in 2014. Available data include questionnaire surveys, physical measures, blood and urine tests, biological samples and genomic data (whole-genome sequencing, whole-genome genotyping, DNA methylation, human leukocyte antigen typing and blood metabolome). Researchers interested in obtaining TWB individual-level data for research purposes would need to submit an application that includes a detailed research proposal and an institutional review board approval to TWB (contact email: biobank@gate.sinica.edu.tw). The application will undergo scientific and ethical reviews by external experts in relevant scientific fields and the TWB ethical governance committee (EGC). Once approved, researchers will be able to access the data for the approved research projects during the approved time period. For international researchers outside of Taiwan, an additional international data transfer agreement needs to be filed to the Ministry of Health and Welfare of Taiwan to enable sharing of the TWB individual-level data and any derived data. Access to KoGES data, including phenotypes and genotypes, is granted upon approval from the Institutional Review Board of the Korean National Institute of Health. Comprehensive details on KoGES data distribution can be found at the Korea Biobank Project

website (https://www.kdca.go.kr/contents.es?mid=a30326000000). Data from the UKBB are available on application to their site (UKBB, https://www.ukbiobank.ac.uk). Data from the NIA-LOAD can be accessed from dbGaP under accession number phs000168.v1.p1. Summary statistics of EUR GWAS for EduYears by Lee and colleagues[14] are publicly available at the Social Science Genetic Association Consortium (SSGAC, https://www.thessgac.org/). The full summary statistics of EAS GWAS and cross-ancestry GWAS are publicly available at the NHGRI-EBI GWAS Catalog (https://www.ebi.ac.uk/gwas/downloads) with accession numbers GCST90296498 and GCST90296499, respectively.

## Code availability

Previously developed pipelines were used to produce the results of the current study. No custom code was developed. Please see Supplementary Information for the list of URLs of the software and data utilized in this study.

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

## Acknowledgements

We thank the participants of TWB and the staff, management team and leadership of TWB. We thank UKBB, dbGaP and the Social Science Genetic Association Consortium for providing resources and releasing the genome-wide association summary statistics that made this study possible. We thank the National Core Facility for Biopharmaceuticals (NCFB, 111-2740-B-492-001) and National Center for High-performance Computing (NCHC) of National Applied Research Laboratories (NARLabs) of Taiwan for providing computational and storage resources. This study was supported by the National Health Research Institutes (NP-110, 111, 112-PP-09 to Y.-F.L.) and the Ministry of Science and Technology, Taiwan (MOST 109-2314-B-400-017 and 110-2314-B-400-028-MY3 to Y.-F.L., 111-2314-B-002-299 and 112-2314-B-002-200-MY3 to Y.-C.A.F., 106-2628-B-010-001-MY4 and 110-2314-B-A49A-524 to Y.-F.C., and 110-2321-B-418-001 to Y.-L.C.). This study was supported by a National Research Foundation of Korea Grant funded by the Ministry of Science and Information and Communication Technologies, South Korea (grant numbers NRF-2021R1A2C4001779 to W.M. and NRF-2022R1A2C2009998 to H.-H.W.), and by the Korea Health Technology R&D Project through the Korea Health Industry Development Institute, funded by the Ministry of Health and Welfare, South Korea (HU22C0042 and HU21C0111 to H.-H.W.). This study was supported by Future Medicine 2030 Project of the Samsung Medical Center (#SMX1230081 to H.-H.W.). This study was conducted using bioresources from the National Biobank of Korea, the Korea Disease Control and Prevention Agency, South Korea (KBN-2021-031). The funders had no role in study design, data collection and analysis, decision to publish or preparation of the manuscript.

## Author contributions

Y.-F.L., W.M. and H.-H.W. have full access to all data in the study and take responsibility for the integrity of the data and accuracy of the data analysis. Y.-F.C. and Y.-L.C. have full access to EMCIT data and take responsibility for the integrity of the data. H.J., H.J.K. and S.W.S. have full access to Korean-based cohort data and take responsibility for the integrity of the data. T.-T.C., J.K., M.L., Y.-F.L., W.M., C.-Y.C. and H.-H.W. conceived and designed the study. T.-T.C., J.K., S.-H.J., B.K., S.K., C.C., I.S., S.P. and Y.A. performed the statistical analyses. T.-T.C., J.K., M.L., Y.-F.L., W.M., C.-Y.C. and H.-H.W. drafted the paper. S.-C.L., A.O., W.-Y.P., Y.-C.A.F., T.G., H.H., Y.-F.L., W.M., C.-Y.C. and H.-H.W. supervised the entire study and critically revised the paper. All authors contributed to interpretation of the data and writing the paper and have read and approved the final draft for submission. T.-T.C. and J.K. contributed equally to this work. Y.-F.L., W.M., C.-Y.C. and H.-H.W. jointly supervised this work.

# Article

## Competing interests

C.-Y.C. is an employee of Biogen. W.-Y.P. is employed by a commercial company, GENINUS. The other authors declare no competing interests.

## Additional information

**Correspondence and requests for materials** should be addressed to Yen-Feng Lin, Woojae Myung, Chia-Yen Chen or Hong-Hee Won.

[1]Center for Neuropsychiatric Research, National Health Research Institutes, Miaoli, Taiwan. [2]Department of Digital Health, Samsung Advanced Institute for Health Sciences and Technology (SAIHST), Sungkyunkwan University, Samsung Medical Center, Seoul, South Korea. [3]Department of Neuropsychiatry, Seoul National University Bundang Hospital, Seongnam, South Korea. [4]Analytic and Translational Genetics Unit, Massachusetts General Hospital, Boston, MA, USA. [5]Department of Medicine, Harvard Medical School, Boston, MA, USA. [6]Human Genetics, Genome Institute of Singapore, Singapore, Singapore. [7]Division of Psychiatry Research, the Zucker Hillside Hospital, Northwell Health, Glen Oaks, NY, USA. [8]Research Division Institute of Mental Health Singapore, Singapore, Singapore. [9]Institute of Public Health and International Health Program, College of Medicine, National Yang Ming Chiao Tung University, Taipei, Taiwan. [10]Graduate Institute of Medicine, Yuan Ze University, Taoyuan City, Taiwan. [11]Department of Medical Research, Far Eastern Memorial Hospital, New Taipei City, Taiwan. [12]Department of Biostatistics, Epidemiology and Informatics, Perelman School of Medicine, University of Pennsylvania, Philadelphia, PA, USA. [13]Stanley Center for Psychiatric Research, the Broad Institute of MIT and Harvard, Cambridge, MA, USA. [14]Department of Economics, School of Business and Economics, Vrije Universiteit Amsterdam, Amsterdam, the Netherlands. [15]Departments of Neurology, Samsung Medical Center, Sungkyunkwan University School of Medicine, Seoul, South Korea. [16]Alzheimer's Disease Convergence Research Center, Samsung Medical Center, Seoul, South Korea. [17]Samsung Genome Institute, Samsung Medical Center, Sungkyunkwan University School of Medicine, Seoul, South Korea. [18]Psychiatric and Neurodevelopmental Genetics Unit, Center for Genomic Medicine, Massachusetts General Hospital, Boston, MA, USA. [19]Department of Psychiatry, Massachusetts General Hospital, Harvard Medical School, Boston, MA, USA. [20]Institute of Health Data Analytics and Statistics, College of Public Health, National Taiwan University, Taipei City, Taiwan. [21]Institute of Epidemiology and Preventive Medicine, College of Public Health, National Taiwan University, Taipei City, Taiwan. [22]Department of Public Health and Medical Humanities, School of Medicine, National Yang Ming Chiao Tung University, Taipei, Taiwan. [23]Institute of Behavioral Medicine, College of Medicine, National Cheng Kung University, Tainan, Taiwan. [24]Department of Psychiatry, Seoul National University College of Medicine, Seoul, South Korea. [25]Biogen, Cambridge, MA, USA. [26]These authors contributed equally: Tzu-Ting Chen, Jaeyoung Kim. [27]These authors jointly supervised this work: Yen-Feng Lin, Woojae Myung, Chia-Yen Chen, Hong-Hee Won. ✉e-mail: yflin@nhri.edu.tw; wmyung@snu.ac.kr; chiayenc@gmail.com; wonhh@skku.edu

# Reporting Summary

## Statistics

For all statistical analyses, confirm that the following items are present in the figure legend, table legend, main text, or Methods section.

| n/a | Confirmed | |
|---|---|---|
| ☐ | ☒ | The exact sample size (*n*) for each experimental group/condition, given as a discrete number and unit of measurement |
| ☐ | ☒ | A statement on whether measurements were taken from distinct samples or whether the same sample was measured repeatedly |
| ☐ | ☒ | The statistical test(s) used AND whether they are one- or two-sided *Only common tests should be described solely by name; describe more complex techniques in the Methods section.* |
| ☐ | ☒ | A description of all covariates tested |
| ☐ | ☒ | A description of any assumptions or corrections, such as tests of normality and adjustment for multiple comparisons |
| ☐ | ☒ | A full description of the statistical parameters including central tendency (e.g. means) or other basic estimates (e.g. regression coefficient) AND variation (e.g. standard deviation) or associated estimates of uncertainty (e.g. confidence intervals) |
| ☐ | ☒ | For null hypothesis testing, the test statistic (e.g. *F*, *t*, *r*) with confidence intervals, effect sizes, degrees of freedom and *P* value noted *Give P values as exact values whenever suitable.* |
| ☐ | ☒ | For Bayesian analysis, information on the choice of priors and Markov chain Monte Carlo settings |
| ☒ | ☐ | For hierarchical and complex designs, identification of the appropriate level for tests and full reporting of outcomes |
| ☐ | ☒ | Estimates of effect sizes (e.g. Cohen's *d*, Pearson's *r*), indicating how they were calculated |

*Our web collection on statistics for biologists contains articles on many of the points above.*

## Software and code

Policy information about availability of computer code

| Data collection | No software was used for data collection. |
|---|---|
| Data analysis | Previously developed pipelines were used to produce the results of the current study. No custom code was developed. LocusZoom v0.9.6, https://my.locuszoom.org/; PBK genotype QC project, https://github.com/Annefeng/PBK-QC-pipeline; UK Biobank quality control documentation, https://www.ukbiobank.ac.uk/wp-content/uploads/2014/04/imputation_documentation_May2015.pdf; LDSC v1.0.1, https://github.com/bulik/ldsc; Regenie v2.2.4, https://rgcgithub.github.io/regenie/; PLINK v2.0,https://www.cog-genomics.org/plink/2.0/; METAL 2020-05-05, https://github.com/statgen/METAL; SnpSift v4_3t_core, https://pcingola.github.io/SnpEff/; S-LDXR v0.3-beta, https://huwenboshi.github.io/s-ldxr/; FUMA v1.3.7, https://fuma.ctglab.nl/; SuSiEx v1.0.0, https://github.com/getian107/SuSiEx/; R v4.2.1, https://www.r-project.org/; susieR package v0.12.10, https://stephenslab.github.io/susieR/index.html; PRS-CS v1.0.0, https://github.com/getian107/PRScs; PRS-CSx v1.0.0, https://github.com/getian107/PRScsx/. |

For manuscripts utilizing custom algorithms or software that are central to the research but not yet described in published literature, software must be made available to editors and reviewers. We strongly encourage code deposition in a community repository (e.g. GitHub). See the Nature Portfolio guidelines for submitting code & software for further information.

# Data

Policy information about availability of data

All manuscripts must include a data availability statement. This statement should provide the following information, where applicable:

- Accession codes, unique identifiers, or web links for publicly available datasets
- A description of any restrictions on data availability
- For clinical datasets or third party data, please ensure that the statement adheres to our policy

A detailed description of the availability and application process of the individual-level TWB data can be found at https://www.biobank.org.tw/english.php. Briefly, TWB made available the individual-level data and biological samples from the participants of the prospective cohort study in 2014. Available data include questionnaire surveys, physical measures, blood and urine tests, biological samples and genomic data (whole-genome sequencing, whole-genome genotyping, DNA methylation, HLA typing, and blood metabolome). Researchers interested in obtaining TWB individual-level data for research purposes would need to submit an application that includes a detailed research proposal and an institutional review board (IRB) approval to TWB (contact email: biobank@gate.sinica.edu.tw). The application will undergo scientific and ethical reviews by external experts in relevant scientific fields and the TWB ethical governance committee (EGC). Once approved, researchers will be able to access the data for the approved research projects during the approved time period. For international researchers outside of Taiwan, an additional international data transfer agreement needs to be filed to the Ministry of Health and Welfare of Taiwan to enable sharing of the TWB individual-level data and any derived data. Access to KoGES data, including phenotypes and genotypes, is granted upon approval from the Institutional Review Board of the Korean National Institute of Health. Comprehensive details on KoGES data distribution can be found at the Korea Biobank Project website (https://www.kdca.go.kr/contents.es?mid=a30326000000). Data from the UKBB is available on application to their site (UK Biobank, https://www.ukbiobank.ac.uk). Data from the NIA-LOAD can be accessed from dbGaP under accession number (phs000168.v1.p1). Summary statistics of EUR GWAS for EduYears by Lee and colleagues are publicly available at the Social Science Genetic Association Consortium (SSGAC, https://www.thessgac.org/). The full summary statistics of EAS GWAS and cross-ancestry GWAS are publicly available at the NHGRI-EBI GWAS Catalog (https://www.ebi.ac.uk/gwas/downloads) with accession numbers GCST90296498 and GCST90296499, respectively.

# Human research participants

Policy information about studies involving human research participants and Sex and Gender in Research.

| | |
|---|---|
| Reporting on sex and gender | Sex was used as a covariate in the GWAS of educational attainment (EduYears) in East Asian. In total, 63,531 males (36%) and 112,879 (64%) females were included in the East Asian GWAS. |
| Population characteristics | For meta-analysis in East Asian population: 104,722 individuals (37,766 males, the birth year ranged from 1939-1989) in Taiwan Biobank (TWB) and 71,678 individuals (25,755 males, the birth year ranged from 1918-1973) in Korean Genome and Epidemiology Study (KoGES). More population characteristics were described in Supplementary Table 1. |
| Recruitment | Researchers in this study were not involved in the participant recruitment. |
| Ethics oversight | This study has been approved by the ethics committee of National Health Research Institutes, Taiwan (TWB; EC1090402-E and EC1110608-E) and Seoul National University Bundang Hospital, South Korea (KoGES; X-2107-699-902). |

Note that full information on the approval of the study protocol must also be provided in the manuscript.

# Field-specific reporting

Please select the one below that is the best fit for your research. If you are not sure, read the appropriate sections before making your selection.

☐ Life sciences    ☒ Behavioural & social sciences    ☐ Ecological, evolutionary & environmental sciences

For a reference copy of the document with all sections, see nature.com/documents/nr-reporting-summary-flat.pdf

# Behavioural & social sciences study design

All studies must disclose on these points even when the disclosure is negative.

| | |
|---|---|
| Study description | This is a genome-wide association study (GWAS) for EduYears in East Asian population, followed by a cross-population GWAS meta-analysis for EduYears between East Asian and European populations. The phenotype data is quantitative. |
| Research sample | The total sample size was 942,745, consisting of 176,400 East Asian samples from two nation-wide biobanks (TWB and KoGES) and 766,345 individuals of European genome-wide summary statistics. Both TWB and KoGES are population-based prospective cohorts by recruiting volunteers. The mean age was 49.9 ± 10.9 years for TWB samples and 54.1 ± 8.3 years for KoGES samples. There were more females than males in both cohorts. Please find the detail information for TWB and KoGES in Supplementary Table 1. In this study, we present the largest-to-date EduYears GWAS in East Asian population and cross-ancestry GWAS meta-analysis across East Asian and European populations for EduYears. |

| | |
|---|---|
| Sampling strategy | Both TWB and KoGES are population-based prospective cohorts by recruiting volunteers. We did not perform sample size calculation; instead, we conducted an international collaborative study to maximize the sample size of East Asian for GWAS on educational attainment. |
| Data collection | In this observational study, data was collected independently in each cohort. Baseline characteristic data were collected from interviews, physical measurements, biomarkers, and genetic data in both TWB and KoGES. Blinding is not applicable as the paper reports a genome-wide association study for an observed variable with no experimental manipulation. |
| Timing | TWB recruited participants from 2012 to 2020, and KoGES recruited participants from 2001 to 2013. |
| Data exclusions | After stringent quality control (QC), we excluded 2,771 samples from TWB and 616 samples from KoGES in the EAS GWAS. |
| Non-participation | No participants dropped out or declined participation. |
| Randomization | This genome-wide association study (GWAS) is an observational study. Randomization is not applicable in this study. |

# Reporting for specific materials, systems and methods

We require information from authors about some types of materials, experimental systems and methods used in many studies. Here, indicate whether each material, system or method listed is relevant to your study. If you are not sure if a list item applies to your research, read the appropriate section before selecting a response.

## Materials & experimental systems

| n/a | Involved in the study |
|---|---|
| ☒ | ☐ Antibodies |
| ☒ | ☐ Eukaryotic cell lines |
| ☒ | ☐ Palaeontology and archaeology |
| ☒ | ☐ Animals and other organisms |
| ☒ | ☐ Clinical data |
| ☒ | ☐ Dual use research of concern |

## Methods

| n/a | Involved in the study |
|---|---|
| ☒ | ☐ ChIP-seq |
| ☒ | ☐ Flow cytometry |
| ☒ | ☐ MRI-based neuroimaging |

