## [Peer Review File · Nature Human Behaviour]

Peer Review Information

Journal: Nature Human Behaviour

Manuscript Title: Shared genetic architectures of educational attainment in East Asian and European populations

Corresponding author name(s): Yen-Feng Lin, Woojae Myung, Chia-Yen Chen, Hong-Hee Won

Reviewer Comments & Decisions:

Decision Letter, initial version:

4th May 2023

Dear Professor Won,

Thank you once again for your manuscript, entitled "Shared genetic architectures of educational attainment in East Asian and European populations", and for your patience during the peer review process.

Your Article has now been evaluated by 3 referees. You will see from their comments copied below that, although they find your work of [considerable] potential interest, they have raised quite substantial concerns. In light of these comments, we cannot accept the manuscript for publication, but would be interested in considering a revised version if you are willing and able to fully address reviewer and editorial concerns.

We hope you will find the referees' comments useful as you decide how to proceed. If you wish to submit a substantially revised manuscript, please bear in mind that we will be reluctant to approach the referees again in the absence of major revisions. We are committed to providing a fair and constructive peer-review process. Do not hesitate to contact us if there are specific requests from the reviewers that you believe are technically impossible or unlikely to yield a meaningful outcome.

Editorially, we ask that you respond to all reviewers' concerns, and implement new analyses suggested by reviewers (including Reviewer #2 point 4, Reviewer #3 Lines 222). We would also ask that you deposit your summary statistics publicly. Please also provide more context for your study, and more fine-grained analyses of regions or pathways that are shared between cohorts (Reviewer #2 point 3, Reviewer #1 point 2). Finally, while this is not a strict requirement, we ask that you take into account the recently released recommendations on using populations descriptors in behavioural genetics, <https://nap.nationalacademies.org/catalog/26902/using-population-descriptors-in-genetics-and-genomics-research-a-new/>

If you wish to submit a suitably revised manuscript, we would hope to receive it within 2 months. I would be grateful if you could contact us as soon as possible if you foresee difficulties with meeting this target resubmission date.

- Include a "Response to the editors and reviewers" document detailing, point-by-point, how you addressed each editor and referee comment. If no action was taken to address a point, you must provide a compelling argument. When formatting this document, please respond to each reviewer comment individually, including the full text of the reviewer comment verbatim followed by your response to the individual point. This response will be used by the editors to evaluate your revision and sent back to the reviewers along with the revised manuscript.
- Highlight all changes made to your manuscript or provide us with a version that tracks changes.

[REDACTED]

Thank you for the opportunity to review your work. Please do not hesitate to contact me if you have any questions or would like to discuss the required revisions further.

Sincerely,

Arunas Radzvilavicius, PhD
Senior Editor, Nature Human Behaviour
Nature Research

Reviewer expertise:

Reviewer #1: statistical genetics, complex phenotypes

Reviewer #2: behavioural genetics

Reviewer #3: behavioural genetics

REVIEWER COMMENTS:

Reviewer #1:

Remarks to the Author:

Authors conducted a GWAS of educational attainment in east Asians, followed by a cross-population GWAS met-analysis with Europeans. They identified shared genetic background and polygenic risk score transferability between east Asians and Europeans. While this manuscript handles potentially interesting topic, this reviewer has concerns.

1. As declared in the manuscript, the east Asian GWAS or cross-population GWAS meta-analysis did not find novel genetic loci associated with educational attainment. This should limit the value of this manuscript.
2. Authors identified polygenic genetic correlations across populations, but it is not surprising for the common traits. Further investigation of which biological pathways or regions had shared or distinct association signals are warranted.
3. There existed an association heterogeneity at the ALDH2 locus between the studies of east Asians. Given highly pleiotropic features of the ALDH2 loci, further investigations to assess the reasons of the association heterogeneity is necessary.
4. The value of the GWAS manuscripts strongly depends on public deposit of the summary statistics without restrictions. Regrettably, authors have no plan to do so, which mitigates the value of this manuscript.

Reviewer #2:

Remarks to the Author:

The authors present the first large-scale GWAS of years of educational attainment among East Asian participants. They also compare and combine those results with the larger previously available set of results from European samples. The paper presents a thoughtful and appropriate analysis pipeline and draws reasonable conclusions from the data. My comments primarily relate to additional context that could be provided considering the general readership of the journal.

1. The "why"

It has been 10 years since the first "successful" GWAS of years of educational attainment. Especially for a general/non-genetics audience, it is necessary to provide a serious discussion of (a) why this analysis and (b) why this phenotype. In particular, there needs to be clear consideration of what, exactly, can and cannot be learned from this approach. For example, in the first paragraph the authors state that because educational attainment is correlated with "various social, economic, and health-related outcomes" and "various diseases, including cardiovascular diseases, metabolic diseases, psychiatric disorders, and Alzheimer's disease" that "research on genetic factors related to EduYears may help to identify modifiable risk factors for various health outcomes." If genes are correlated with

educational attainment and educational attainment is correlated with health, how does knowing what the genes are help to "identify modifiable risk factors"? (Counter examples might include: we didn't need to know the gene for PKU to develop a test and dietary intervention; we still don't understand the genetic architecture of most forms of myopia, but glasses work.) These sorts of broad general claims about implications for prediction or intervention are made several times throughout the manuscript and should be substantially more thoroughly developed to not be open to misinterpretation by either a general science audience or the public.

2. Cross-ancestry r_g

For me, the most interesting part of the report is the cross-ancestry r_g , which was 0.87 *both* between cohorts within the EAS sample and between the EAS and EUR results. It would be helpful for the general readership to contextualize this by comparing it to other cross-ancestry genetic correlations that have been reported. I think this result is worth greater emphasis and explanation. (The other result I found fascinating was the drop in the PGS r^2 applied to the UK-based Chinese sample, for the potential implied gene-environment interplay as mentioned by the authors on page 17, although further evaluation of this effect may be outside the scope of the current report.)

3. Post-processing of GWAS results

Also following from the high r_g , I think the discussion and comparison between EAS and EUR samples on the post-processing results is of relatively low interest. Given the high r_g between the EAS and EUR samples, it is essentially a given that the identified systems and cross-phenotype genetic correlations will be substantially the same, but this does not provide independent information about architecture/overlap. For example, the statement on page 15, "we found that EduYears-associated variants were enriched in the CNS in both EAS and EUR populations... suggesting that the CNS is important for EduYears" is in some ways frivolous, in that it is true for all psychologically relevant traits. It may be helpful to contextualize these results in terms of what possible genetic architectures specifically have been supported versus ruled out by each of these analyses.

4. Possible "all else equal..." comparisons

The authors show that combining the total current EAS and EUR results don't identify novel variants, but this is not surprising because there is a substantial discrepancy in N between the EAS and EUR GWAS samples. An interesting question, although I admit it to perhaps be outside of the scope of the current report, would be how consistent the results are when they represent a mixture of EAS and EUR versus EAS only or EUR only *while holding the total N constant*. That is, does performance improve with a mix of (for example) 50k EAS + 50k EUR versus 100k EAS *or* EUR? Although this would use a subset of the currently available data, the illustration would be useful for informing participant recruitment and data analysis decisions for researchers going forward.

I sign my reviews.
Jaime Derringer

Reviewer #3:

Remarks to the Author:

This is a genome-wide association study (GWAS) of years of education (EduYears). Previous GWAS of this trait in Europeans (EUR) have been published in prominent outlets (Science, Nature, Nature Genetics) and garnered thousands of citations. The GWAS reported in the present paper was of two East Asian (EAS) populations. The sample size ($\sim 170,000$) was not enough to yield results comparable to those from the GWAS of EUR (latest sample size ~ 3 million), but nevertheless produced very strong and valuable findings. The genetic architecture of EduYears is very similar in EUR and EAS. Genetic correlations with other traits are correspondingly very similar in both populations, as are inferences from biological annotation. The EAS GWAS was shown to be useful in improving the fine-mapping of causal variants.

This is an excellent paper that should be published. I do not have any major suggestions. Here are some minor suggestions and questions (most of which need not be addressed in order to secure my assent to acceptance of the manuscript):

Lines 172-173: Supplementary Table 6 suggests that the additional baseline annotations introduced in Gazal et al. (2017) were not used. The impact of using these will probably be to shrink the enrichment estimates toward one, particularly that of conserved_Lindblad. The overall interpretation will probably be scarcely affected, but the authors might want to consider the update (and perhaps any later ones that I'm not aware of).

Lines 219-221: Intuitively the EUR-specific meta-analysis should yield more hits with the help of the EAS results than without. Is it possible to provide some numbers to back up this intuition? How many hits come out of the GWAS of $\sim 760K$ Europeans when MAMA is not used to bring in the EAS data?

Lines 222-235: Some readers might not be impressed by the concordance between EAS and EUR at these top SNPs. Supplementary 12 makes it clear that the estimates in EAS are much more perturbed by sampling error. It might help to use a more formal framework incorporating variation in statistical power to derive benchmarks for comparison, such as the framework used by Okbay et al. (2022, SI Section 2.3)

Lines 346-347: I suggest giving only the HUGO gene symbol for FAM81B rather than the full name that the symbol stands for.

Tables 1 and 2: What does "credible set ID" mean? What does it mean for a gene to be listed (or not) for a particular row? The table notes should explain these.

Supplementary Figure 5: This scatterplot is pretty useful. But even more useful might be a calculation of F_{ST} between the two EAS populations, either from the data in this study or in another publication. F_{ST} between the extremes of northern and southern Europe is about 0.01, and I believe it is common practice to include cohorts from all latitudes of Europe in EUR meta-analyses.

Supplementary Tables 7-9: Why not include the LDSC-SEG enrichments as well as the coefficients?

Supplementary Table 15: The source of the GWAS summary statistics is sometimes given as "Figshare." Is this a mistake?

REFERENCES

Gazal, S. et al. (2017). Linkage disequilibrium-dependent architecture of human complex traits shows action of negative selection. *Nature Genetics*, 49, 1421-1427.

Okbay, A. et al. (2022). Polygenic prediction of educational attainment within and between families from genome-wide association analyses in 3 million individuals. *Nature Genetics*, 54, 437-449.

Author Rebuttal to Initial comments

Responses from the Authors to Review Comments:

We are grateful to the reviewers for their valuable feedback and insightful comments that helped us to improve the quality of our manuscript. We have carefully considered all the comments and revised the manuscript accordingly. Our point-wise responses to the reviewers' comments are given below.

REVIEWER COMMENTS:

Reviewer #1:

Remarks to the Author:

Authors conducted a GWAS of educational attainment in east Asians, followed by a cross-population GWAS meta-analysis with Europeans. They identified shared genetic background and polygenic risk score transferability between east Asians and Europeans. While this manuscript handles potentially interesting topic, this reviewer has concerns.

1. As declared in the manuscript, the east Asian GWAS or cross-population GWAS meta-analysis did not find novel genetic loci associated with educational attainment. This should limit the value of this manuscript.

Response: We really appreciate the valuable comments provided by the reviewer. We acknowledge that our study did not identify novel genetic loci for educational attainment in the EAS GWAS compared with previous educational attainment GWAS in EUR. However, we believe that our research holds significant value within the field of social and behavioural genetics, particularly with respect to exploring the shared genetic architectures between ethnically and socially diverse populations. As highlighted by Martin et al. (Ref 1), there exists a notable imbalance in the representation of participants in GWAS, with a disproportionate focus on European populations compared to non-European populations. Given this limitation, our cross-ancestry GWAS analysis presents a timely and relevant contribution to the field. By explicitly examining the shared genetic foundations of educational attainment across different ethnic groups, we aim to provide a more comprehensive understanding of the genetic factors that influence educational attainment.

Accordingly, we have added limitation of the study regarding the absence of novel genetic loci associated with educational attainment.

Ref 1. Martin, Alicia R., et al. "Clinical use of current polygenic risk scores may exacerbate health disparities." *Nature genetics* 51.4 (2019): 584-591.

[Added to the Discussion, pages 20–21, lines 463–471]

However, compared to the largest GWAS for *EduYears* in EUR, considerably fewer genomic loci in EAS were identified (7 loci in EAS vs. 3,952 loci in EUR) and all seven loci reported in EAS were previously reported in the EUR GWAS. The absence of novel loci in the *EduYears* GWAS in EAS compared with previous EUR GWAS reflects the lower power for gene discovery with the current sample size in TWB and KoGES. However, we expect to obtain more insight into the genetic basis of *EduYears* in the EAS population as the sample size increases with more samples from TWB and KoGES, as well as the inclusion of more EAS cohorts.

2. Authors identified polygenic genetic correlations across populations, but it is not surprising for the common traits. Further investigation of which biological pathways or regions had shared or distinct association signals are warranted.

Response: We appreciate the reviewer's insightful comment regarding the investigation of shared or distinct association signals across biological pathways or regions among different populations. In response to this suggestion, we utilised the GSA-SNP2 tool (Genetic Set Association of SNP Prioritization 2) to perform a comparative analysis of biological pathways between the two populations. The outcomes of this analysis highlighted substantial similarities in the association patterns between *EduYears* and pathways across populations (**Fig. 2c**). This observation indicates that certain biological pathways are consistently implicated in the genetic basis of educational attainment, regardless of ancestral backgrounds.

[Added to the Results, page 11, lines 227–236]

Finally, we conducted pathway enrichment analysis using the Gene Set Analysis-Single-Nucleotide-Polymorphism-2 (GSA-SNP2)²⁹ to explore potential biological

pathways associated with *EduYears*. Based on the GWAS summary statistics from EAS and EUR populations, we aimed to identify pathways significantly associated with *EduYears* in each population and subsequently compare the results to determine shared or distinct pathways between two populations. In total, 16 and 27 pathways were identified as significantly associated with *EduYears* in EAS and EUR populations, respectively (**Fig. 2c**). Among these significantly enriched pathways, 14 pathways were common across both populations, while two and 13 pathways exhibited significant enrichment exclusively in EAS and EUR populations, respectively.

[Added to the Discussion, pages 17–18, lines 389–395]

Additionally, the pathway enrichment analysis demonstrated shared biological pathways between the EAS and EUR populations. We showed that 14 pathways were significantly associated with *EduYears* in both populations. These findings suggest the consistent involvement of specific biological pathways in the genetic basis of educational attainment, regardless of ancestry. Furthermore, these shared pathways underscore their potential importance in contributing to the association with educational attainment across diverse populations.

[Added to the Methods, page 30, lines 696–705]

Pathway enrichment analyses in EAS and EUR populations

We applied GSA-SNP2²⁹ based on all *P*-values from both EAS and EUR GWAS to detect biological pathways associated with *EduYears*. GSA-SNP2 employs the Z-statistics of the random set model, assessing pathways by combining adjusted gene scores for SNP counts in each gene using a monotone cubic spline trend curve. We evaluated gene set enrichment using the MSigDB C5 collection v5.2 database^{69,70}. For the detailed options regarding the genes and

pathways in the analysis, the race was selected as ancestry-matched (European or East Asian), the reference genome version was set as GRCh37 (hg19), the padding size for genes was set to 20 kb, and the pathway size window was chosen as 10–200. Significantly enriched pathways were defined as those with q -value < 0.05 .

3. There existed an association heterogeneity at the *ALDH2* locus between the studies of east Asians. Given highly pleiotropic features of the *ALDH2* loci, further investigations to assess the reasons of the association heterogeneity is necessary.

Response: We greatly appreciate this insightful comment from the reviewer. We agree with the reviewer's comment that a more in-depth investigation is necessary to comprehend the potential association heterogeneity within the East Asian population. Therefore, we directed our focus towards the *ALDH2* region, which exhibited significant heterogeneity (P -value $< 5 \times 10^{-8}$) in our East Asian-specific GWAS result.

Firstly, we calculated fixation index (F_{st}) values between TWB and KoGES for variants located in the *ALDH2* region. The obtained F_{st} values ranged from 4.12×10^{-8} to 0.045. When mapping the F_{st} values into percentiles across all loci that were included in EAS GWAS meta-analysis, 41 out of 66 variants (62.1%) exhibited percentiles of 50% or higher. This finding suggests the presence of distinctive genetic structures within the *ALDH2* region, indicating potential genetic dissimilarities between the TWB and KoGES.

Subsequently, we examined the phenome-wide association study (PheWAS) results for the *ALDH2* region in KoGES. Notably, the analysis revealed that total alcohol consumption

exhibited the most significant association (**Supplementary Table 5**). Based on this result, we estimated genetic correlation between alcohol drinking and *EduYears* within KoGES, leading to the identification of a significant negative genetic correlation ($r_g = -0.193$; s.e. = 0.063; P -value = 0.002). In contrast, we did not observe any significant genetic correlation between alcohol drinking and *EduYears* within TWB ($r_g = 0.058$; s.e. = 0.299; P -value = 0.8465).

Moreover, we estimated local genetic correlations within KoGES, and we observed the *ALDH2* region showed significant correlation ($\rho = -0.82$, P -value = 7.4×10^{-6}), approximately four times higher than the global genetic correlation between alcohol drinking and *EduYears*.

Lastly, we conducted a stratified GWAS for *EduYears* based on alcohol drinking status within KoGES. In the drinker group, the *ALDH2* region showed a significant association with *EduYears*, while in the non-drinker group, the association was not significant (**Supplementary Fig. 7**). Motivated by these findings, we further explored gene-environment (G×E) interactions within KoGES, considering individuals without stratifying them based on alcohol drinking status. Notably, our analysis unveiled significant G×E interactions within the *ALDH2* region, suggesting potential interactions between these genetic variants and alcohol consumption, affecting educational attainment within the Korean population. We attempted to conduct a similar stratified GWAS and interaction analysis for *EduYears* based on alcohol drinking status in TWB; however, there was no significant result for *EduYears* and the interaction between *EduYears* and alcohol drinking in TWB.

Collectively, our findings suggest that the observed heterogeneity between TWB and KoGES could be attributed to gene-environment interactions influenced by alcohol drinking in KoGES and highlight the importance of studying ethnically and socially diverse populations.

[Added to the Results, page 9, lines 170–187]

Heterogeneity of genetic effects within EAS population

Given that the *ALDH2* region on chromosome 12 showed a significant association with *EduYears* exclusively in KoGES but not in TWB, we conducted further investigation to explore potential underlying factors driving this observed heterogeneity. Firstly, we examined the phenome-wide association study (PheWAS) results for the *ALDH2* region in KoGES and demonstrated that total alcohol consumption exhibited the most significant association with this locus²⁰ (**Supplementary Table 5**). Based on this finding, we estimated the genetic correlation (r_g) between alcohol drinking and *EduYears* in KoGES, both globally and locally. We identified a significant negative global genetic correlation between alcohol drinking and *EduYears* ($r_g = -0.193$; s.e. = 0.063; P -value = 0.002). Moreover, specifically within the *ALDH2* region, we observed a substantial local genetic correlation ($\rho = -0.82$, P -value = 7.4×10^{-6}). In addition, we conducted a stratified GWAS for *EduYears*, segregating KoGES participants into groups of drinkers and non-drinkers. Remarkably, in the drinker group, the *ALDH2* region displayed a significant association with *EduYears* (P -value = 2.4×10^{-22}), while in the non-drinker group, the association was not significant (P -value = 0.032) (**Supplementary Fig. 7**). These findings suggest that the observed heterogeneity in the *ALDH2* region is likely attributed to potential shared genetic component and gene-environment interactions between alcohol drinking and *EduYears*, particularly in KoGES.

[Added to the Discussion, page 17, lines 378–382]

We have confirmed that the observed heterogeneity in the *ALDH2* region may be linked to possible shared genetic component and gene-environment interaction between alcohol drinking

and *EduYears*, in the Korean population. This finding suggests that studying diverse populations can bring new insights in identifying gene-environment associations.

[Added to the Method, page 28, lines 640–656]

Heterogeneity of genetic effects within EAS population

To identify underlying factors contributing to heterogeneity between TWB and KoGES, we performed the following procedures.

PheWAS lookup: To investigate the pleiotropic effects of variants showing heterogeneity, we conducted a search in the KoGES PheWeb (see URLs).

Global and local genetic correlation analyses: To further explore the relationship between alcohol-related traits and *EduYears*, we performed global and local genetic correlation analyses using KoGES data. The global genetic correlation was estimated using LDSC v1.0.1¹⁹, while the local genetic correlation within specific genomic regions was assessed using LAVA⁶⁸. The details regarding the LAVA analysis are summarised in the **Supplementary Note**.

Stratified GWAS analysis: In KoGES, individuals with a history of past alcohol consumption or those currently engaged in alcohol consumption were categorized into the drinker group, while individuals with no history of alcohol consumption were assigned to the non-drinker group. We then performed genetic association analyses for *EduYears* using Regenie v2.2.4⁶³, adjusting for BY, BY², BY³, sex, BY by sex interaction, BY² by sex interaction, BY³ by sex interaction, and the top ten PCs.

4. The value of the GWAS manuscripts strongly depends on public deposit of the summary statistics without restrictions. Regrettably, authors have no plan to do so, which mitigates the value of this manuscript.

Response: We express our gratitude to the reviewer for this comment emphasising the importance of public deposition of summary statistics. We understand the value of data sharing in promoting transparency, reproducibility, and collaboration in the scientific community. We are committed to depositing the summary statistics derived from our GWAS analysis into the GWAS Catalog once our manuscript is accepted. This proactive step will enable fellow researchers to readily access and utilise the data for subsequent exploration and replication endeavours.

[Added to the Data availability, page 37, lines 849–850]

The full summary statistics of GWAS in EAS is publicly available at the GWAS Catalog (<https://www.ebi.ac.uk/gwas>).

We would like to express our gratitude for the valuable comments and suggestions. We really appreciate the time and effort invested in reviewing our manuscript.

Reviewer #2:

Remarks to the Author:

The authors present the first large-scale GWAS of years of educational attainment among East Asian participants. They also compare and combine those results with the larger previously available set of results from European samples. The paper presents a thoughtful and appropriate analysis pipeline and draws reasonable conclusions from the data. My comments primarily relate to additional context that could be provided considering the general readership of the journal.

1. The "why"

It has been 10 years since the first "successful" GWAS of years of educational attainment. Especially for a general/non-genetics audience, it is necessary to provide a serious discussion of (a) why this analysis and (b) why this phenotype. In particular, there needs to be clear consideration of what, exactly, can and cannot be learned from this approach. For example, in the first paragraph the authors state that because educational attainment is correlated with "various social, economic, and health-related outcomes" and "various diseases, including cardiovascular diseases, metabolic diseases, psychiatric disorders, and Alzheimer's disease" that "research on genetic factors related to EduYears may help to identify modifiable risk factors for various health outcomes." If genes are correlated with educational attainment and educational attainment is correlated with health, how does knowing what the genes are help to "identify modifiable risk factors"? (Counter examples might include: we didn't need to know the gene for PKU to develop a test and dietary intervention; we still don't understand the genetic architecture of most forms of myopia, but glasses work.) These sorts of broad general claims about implications for prediction or intervention are made several times throughout the manuscript and should be substantially more thoroughly developed to not be open to misinterpretation by either a general science audience or the public.

Response: We really appreciate the valuable comments from the reviewer. We agree on the reviewer's suggestion and have thoroughly revised both the Introduction and Discussion sections of the manuscript. We highlight the merits of broadening the scope of educational attainment GWAS to encompass East Asian populations, thereby enriching our comprehension of the genetic basis of *EduYears* and facilitating the transferability of genetic insights across diverse populations. Moreover, we have taken great care to enhance the introductory context, providing a more appropriate rationale for the execution of this study. Furthermore, a dedicated discussion has been included to meticulously explore the advantages and limitations of studying *EduYears* as a proxy phenotype for other health-related outcomes.

2. Cross-ancestry r_g

For me, the most interesting part of the report is the cross-ancestry r_g , which was 0.87 *both* between cohorts within the EAS sample and between the EAS and EUR results. It would be helpful for the general readership to contextualize this by comparing it to other cross-ancestry genetic correlations that have been reported. I think this result is worth greater emphasis and explanation. (The other result I found fascinating was the drop in the PGS r^2 applied to the UK-based Chinese sample, for the potential implied gene-environment interplay as mentioned by the authors on page 17, although further evaluation of this effect may be outside the scope of the current report.)

Response: We express our sincere gratitude for the valuable comment provided by the reviewer. We have included additional discussion to encompass the level of genetic correlation within EAS populations and between EAS and EUR populations and a comparison with cross-population genetic correlation for other traits.

[Added to the Discussion, pages 16–17, lines 359–371]

This study provides several novel findings regarding the genetics of *EduYears*. First, we observed high positive genetic correlations of *EduYears* within the EAS population ($r_g = 0.87$) and between the EAS and EUR populations ($r_g = 0.87$). This suggests a comparable degree of shared genetic component for *EduYears* within the EAS and between EAS and EUR. To benchmark the EAS-EUR cross-population r_g for *EduYears* against other traits, we extracted EAS-EUR cross-population r_g for 31 other traits from Shi *et al.*³² as a reference. Remarkably, the cross-population r_g for *EduYears* closely aligns with the median of EAS-EUR cross-population r_g across the 31 traits (median $r_g = 0.88$; range = 0.342 to 1.05). While the EAS-EUR cross-population r_g for *EduYears* is lower than that for schizophrenia (EAS-EUR cross-population $r_g = 0.945$), it is considerably higher than major depressive disorder (EAS-EUR cross-population $r_g = 0.342$) and comparable to other physiological traits (EAS-EUR cross-population $r_g = 0.897$ for height) and molecular phenotypes (EAS-EUR cross-population $r_g = 0.875$ for hemoglobin A1c).

3. Post-processing of GWAS results

Also following from the high r_g , I think the discussion and comparison between EAS and EUR samples on the post-processing results is of relatively low interest. Given the high r_g between the EAS and EUR samples, it is essentially a given that the identified systems and cross-phenotype genetic correlations will be substantially the same, but this does not provide independent information about architecture/overlap. For example, the statement on page 15, "we found that *EduYears*-associated variants were enriched in the CNS in both EAS and EUR populations... suggesting that the CNS is important for *EduYears*" is in some ways frivolous, in that it is true for all psychologically relevant traits. It may be helpful to contextualize these results in terms of what possible genetic architectures specifically have been supported versus ruled out by each of these analyses.

Response: We appreciate the valuable feedback from both Reviewer 1 and this reviewer. Considering their comments, we agree that the discussion and comparison between the EAS and EUR samples regarding post-processing results may be of relatively limited interest due to the high genetic correlation observed between these populations. While it is expected that the identified systems and cross-phenotype genetic correlations would be similar. We acknowledge that this does not provide independent information regarding the genetic architecture or overlap. In light of this feedback, we have significantly reduced the relevant section. We recognize the valid concern raised regarding the enrichment of *EduYears*-associated variants in the CNS in both EAS and EUR populations. We acknowledge that this finding may not provide specific insights into the genetic architecture of educational attainment, regardless the fact that it identifies psychologically relevant features in general. In addition, we have addressed the suggestions from both this reviewer and Reviewer 1. Specifically, we employed the GSA-SNP2 tool (Genetic Set Association of SNP Prioritization 2) to compare biological pathways between the two populations. Through this tool, we aimed to gain further insights into the shared and distinct genetic signals underlying educational attainment. Our analysis using the GSA-SNP2 tool revealed intriguing results. We observed substantial similarities in the association patterns between *EduYears* and pathways across populations (**Fig. 2c**), indicating a consistent involvement of specific biological pathways in the genetic basis of educational attainment in both ancestries.

[Added to the Results, page 11, lines 227–236]

Finally, we conducted pathway enrichment analysis using the Gene Set Analysis-Single-Nucleotide-Polymorphism-2 (GSA-SNP2)²⁹ to explore potential biological pathways associated with *EduYears*. Based on the GWAS summary statistics from EAS and EUR

populations, we aimed to identify pathways significantly associated with *EduYears* in each population and subsequently compare the results to determine shared or distinct pathways between two populations. In total, 16 and 27 pathways were identified as significantly associated with *EduYears* in EAS and EUR populations, respectively (**Fig. 2c**). Among these significantly enriched pathways, 14 pathways were common across both populations, while two and 13 pathways exhibited significant enrichment exclusively in EAS and EUR populations, respectively.

[Added to the Discussion, pages 17–18, lines 389–395]

Additionally, the pathway enrichment analysis demonstrated shared biological pathways between the EAS and EUR populations. We showed that 14 pathways were significantly associated with *EduYears* in both populations. These findings suggest the consistent involvement of specific biological pathways in the genetic basis of educational attainment, regardless of ancestry. Furthermore, these shared pathways underscore their potential importance in contributing to the association with educational attainment across diverse populations.

[Added to the Methods, page 30, lines 696–705]

Pathway enrichment analyses in EAS and EUR populations

We applied GSA-SNP2²⁹ based on all *P*-values from both EAS and EUR GWAS to detect biological pathways associated with *EduYears*. GSA-SNP2 employs the Z-statistics of the random set model, assessing pathways by combining adjusted gene scores for SNP counts in each gene using a monotone cubic spline trend curve. We evaluated gene set enrichment using the MSigDB C5 collection v5.2 database^{69,70}. For the detailed options regarding the genes and pathways in the analysis, the race was selected as ancestry-matched (European or East Asian),

the reference genome version was set as GRCh37 (hg19), the padding size for genes was set to 20 kb, and the pathway size window was chosen as 10–200. Significantly enriched pathways were defined as those with q -value < 0.05 .

4. Possible "all else equal..." comparisons

The authors show that combining the total current EAS and EUR results don't identify novel variants, but this is not surprising because there is a substantial discrepancy in N between the EAS and EUR GWAS samples. An interesting question, although I admit it to perhaps be outside of the scope of the current report, would be how consistent the results are when they represent a mixture of EAS and EUR versus EAS only or EUR only *while holding the total N constant*. That is, does performance improve with a mix of (for example) 50k EAS + 50k EUR versus 100k EAS *or* EUR? Although this would use a subset of the currently available data, the illustration would be useful for informing participant recruitment and data analysis decisions for researchers going forward.

Response: We believe that the analysis suggested by the reviewer holds important implications. While there are limitations due to the sample size discrepancy between EAS and EUR populations, we have incorporated the reviewer's suggestion and conducted GWAS analyses on five different sets, as described below:

- (1) 176,400 cross-ancestry GWAS: 88,200 EAS (TWB) + 88,200 EUR (UKBB)
- (2) 176,400 EAS (TWB+KoGES)
- (3) 176,400 EUR (UKBB)
- (4) 352,800 cross-ancestry GWAS: 176,400 EAS (TWB+KoGES) + 176,400 EUR (UKBB)

(5) 352,800 EUR (UKBB)

The results are as follows:

1. Results from each GWAS analysis for *EduYears*

Population	N = 88,200	N = 176,400	N = 352,800
	#sig. SNP	#sig. SNP	#sig. SNP
EAS	1	7	NA
EUR (UKBB)	10	59	209
EAS + EUR (UKBB)	NA	14	84

2. Results from polygenic prediction of *EduYears* with fixed sample size (**Supplementary Fig. 20 and Supplementary Table 20**)

Testing cohorts	PRS-CS EAS		PRS-CS EUR (UKBB)		PRS-CSx EAS+EUR (UKBB)	
	(N = 176,400)		(N = 176,400)		(N = 176,400)	
	R ²	P -value	R ²	P -value	R ²	P -value
EMCIT	0.0152	0.0141	0.0031	0.2653	0.006	0.1218
Korean-based	0.034	3.80E-19	0.01	1.42E-06	0.0184	5.37E-11
UKBB (China)	0.0118	5.10E-06	0.0057	0.0015	0.0062	0.0001
NIA-LOAD	0.0046	0.016	0.0438	8.88E-14	0.0123	9.00E-05

Testing cohorts	PRS-CS EAS		PRS-CS EUR (UKBB)		PRS-CSx EAS+EUR (UKBB)	
	(N = 352,800)		(N = 352,800)		(N = 352,800)	
	R ²	P -value	R ²	P -value	R ²	P -value
EMCIT			0.01	0.0464	0.016	0.013
Korean-based			0.0139	1.30E-08	0.0399	2.58E-22

UKBB (China)		0.0102	2.21E-05	0.0166	6.41E-08
NIA-LOAD		0.06	1.90E-18	0.0517	4.87E-16

Overall, we found that GWAS from a single ancestry identified more GWAS loci compared to the meta-analysis of both ancestries. However, PGS prediction using GWAS results from both ancestries, exhibited enhanced explanatory power for educational attainment compared to PGS generated solely from a single ancestry in EAS target cohorts.

These findings align with previous research, such as Graham et al. for lipid study (Ref 1) and Lam et al. for schizophrenia study (Ref 2). These studies similarly demonstrated that PGS based on GWAS results from multiple ancestries yielded better predictive performance. Although our cross-ancestry GWAS analysis did not identify more GWAS loci in the meta-analysis, our fine-mapping results demonstrated the potential to identify more credible causal SNPs.

Accordingly, we have included these results and related discussions in the revised manuscript.

Ref 1. Graham, Sarah E., et al. "The power of genetic diversity in genome-wide association studies of lipids." *Nature* 600.7890 (2021): 675-679.

Ref 2. Lam, Max, et al. "Comparative genetic architectures of schizophrenia in East Asian and European populations." *Nature genetics* 51.12 (2019): 1670-1678.

Furthermore, we recently re-conducted a sample-level quality control (QC) for the NIA-LOAD dataset. As a result, ten additional samples were excluded during the QC process. All the analyses mentioned above were performed using the newly QC'd dataset. To maintain consistency, we also conducted our primary analysis again using the updated QC'd dataset. Consequently, minor adjustments were made to **Fig. 4, Supplementary Tables 1 and 19**.

[Added to the Results, pages 15–16, lines 340–346]

To investigate whether the improvement in predictive performance in the cross-population PGS was solely attributed to an increase in sample size or also influenced by ancestral diversity, we conducted an additional analysis by equating the sample sizes of EAS and EUR populations. Consistent with previous results, the cross-population PGS explained a greater proportion of phenotypic variance in *EduYears* than the EUR-derived PGS in the EAS cohorts (**Supplementary Fig. 20 and Supplementary Table 20**).

[Added to the Discussion, page 20, lines 444–449]

Even with the same sample size, the cross-population PGS consistently outperformed the EUR-derived PGS in the EAS testing cohorts. This observation suggests that population diversity enhanced predictive performance. Through PGS analyses, we explored the transferability of PGS between EAS and EUR populations, which is critical information

regarding the utility of PGS. Furthermore, our PGS analyses also indicated the advantages of ancestral diversity over a single population in PGS construction³⁷.

[Added to the Method, page 34, lines 797–798]

The details for the analyses under the same sample sizes are summarised in the **Supplementary Note**.

We would like to express our gratitude for the valuable comments and suggestions. We really appreciate the time and effort invested in reviewing our manuscript.

Reviewer #3:

Remarks to the Author:

This is a genome-wide association study (GWAS) of years of education (EduYears). Previous GWAS of this trait in Europeans (EUR) have been published in prominent outlets (Science, Nature, Nature Genetics) and garnered thousands of citations. The GWAS reported in the present paper was of two East Asian (EAS) populations. The sample size (~170,000) was not enough to yield results comparable to those from the GWAS of EUR (latest sample size ~3 million), but nevertheless produced very strong and valuable findings. The genetic architecture of EduYears is very similar in EUR and EAS. Genetic correlations with other traits are correspondingly very similar in both populations, as are inferences from biological annotation. The EAS GWAS was

shown to be useful in improving the fine-mapping of causal variants.

This is an excellent paper that should be published. I do not have any major suggestions. Here are some minor suggestions and questions (most of which need not be addressed in order to secure my assent to acceptance of the manuscript):

1. Lines 172-173: Supplementary Table 6 suggests that the additional baseline annotations introduced in Gazal et al. (2017) were not used. The impact of using these will probably be to shrink the enrichment estimates toward one, particularly that of conserved_Lindblad. The overall interpretation will probably be scarcely affected, but the authors might want to consider the update (and perhaps any later ones that I'm not aware of).

Response: We appreciate the reviewer's valuable comments. Accordingly, we have included the results obtained by incorporating the Gazal annotation and updated them accordingly in the revised manuscript, which significantly enhanced the comprehensiveness of our analyses and reflected the most updated information.

[Added the Results, page 10, lines 200–210]

Second, we employed a stratified LDSC^{25,26} with 97 baseline-LD²⁷ for our EAS GWAS summary statistics and EUR summary statistics by Lee *et al.*¹⁵. Among the 97 stratified LDSC annotations, we observed significant enrichments for *EduYears* in the EAS population in six annotations, including H3K4me1 peaks (false discovery rate [FDR] < 5%; **Supplementary Fig. 8** and **Supplementary Table 8**). In the EUR population, 17 annotations, including the conserved primate phastCons46way annotation, representing genomic regions conserved across primate

species, showed significant enrichment for *EduYears* (false discovery rate [FDR] < 5%; **Supplementary Table 9**). Furthermore, ten MAF binary annotations were included to model MAF-dependent architectures within the set of 97 annotations. Of these ten MAF bins, five (more common MAF bins) exhibited significant enrichments for *EduYears* in both EAS and EUR populations.

[Added to the Discussion, page 17, lines 387–389]

Indeed, consistent with the high genetic correlation and transferability observed between EAS and EUR populations, our partitioned heritability and LDSC-SEG analyses^{25,26} showed similar results for both populations.

[Revised the Method, pages 29–30, lines 688–694]

Partitioned heritability analysis

Based on GWAS summary statistics of EAS samples, we used LDSC-SEG v1.0.1²⁶ to prioritise tissues and cell types relevant to *EduYears*. We partitioned genome-wide SNP heritability into 97 baseline-LD annotations introduced by Gazal *et al.*²⁷ and nine tissue-specific categories as specified by Finucane *et al.*²⁵. We used LD scores for the EAS and EUR populations using the 1KG Project phase 3 data provided by LDSC GitHub repository as a reference (see URLs).

2. Lines 219-221: Intuitively the EUR-specific meta-analysis should yield more hits with the help of the EAS results than without. Is it possible to provide some numbers to back up this intuition? How many hits come out of the GWAS of ~760K Europeans when MAMA is not used to bring in the EAS data?

Response: In response to your insightful comments, we have analysed the data using MAMA to incorporate the EAS data and the EUR data. Contrary to our initial expectations, the incorporation of EAS data did not lead to a significant increase in the number of hits when incorporating EAS data (see below table).

Population	N	Lead SNPs	Lead SNPs
		(Before MAMA)	(After MAMA)
EAS	176,400	11	94
EUR	766,345	766	357

This observation can be attributed to the relatively smaller sample size of the EAS population compared to EUR, along with potential heterogeneity between these two populations.

Furthermore, as suggested by Reviewer 2, we have conducted meta-analyses using various sample sizes through the METAL software. Corresponding to the MAMA results, our findings demonstrate that the cross-ancestry GWAS meta-analysis between EUR and EAS did not yield more hits than conducting separate analyses within each ancestry.

Population	N = 88,200	N = 176,400	N = 352,800
	#sig. SNP	#sig. SNP	#sig. SNP
EAS	1	7	NA
EUR (UKB)	10	59	209
EAS + EUR (UKB)	NA	14	84

However, as demonstrated in the PGS and fine-mapping analyses, the cross-ancestry GWAS meta-analysis showed improved predictive power in predicting outcomes in different ancestral populations. Furthermore, this approach aided in identifying credible causal variants compared to single-ancestry GWAS. Consistent with previous studies (Ref 1, Ref 2), our findings indicate that GWAS meta-analysis strengthens predictive ability and fine-mapping across diverse ancestral backgrounds, rather than merely identifying additional GWAS loci.

Ref 1. Graham, Sarah E., et al. "The power of genetic diversity in genome-wide association studies of lipids." *Nature* 600.7890 (2021): 675-679.

Ref 2. Lam, Max, et al. "Comparative genetic architectures of schizophrenia in East Asian and European populations." *Nature genetics* 51.12 (2019): 1670-1678.

We have revised the manuscript to include these insights in response to the reviewer's comments. Thank you for bringing up this point, and we appreciate the opportunity to provide further clarification on the results.

[Added to the Results, pages 15–16, lines 340–346]

To investigate whether the improvement in predictive performance in the cross-population PGS was solely attributed to an increase in sample size or also influenced by ancestral diversity, we conducted an additional analysis by equating the sample sizes of EAS and EUR populations. Consistent with previous results, the cross-population PGS explained a greater proportion of phenotypic variance in *EduYears* than the EUR-derived PGS in the EAS cohorts (**Supplementary Fig. 20** and **Supplementary Table 20**).

[Added to the Discussion, page 20, lines 444–449]

Even with the same sample size, the cross-population PGS consistently outperformed the EUR-derived PGS in the EAS testing cohorts. This observation suggests that population diversity enhanced predictive performance. Through PGS analyses, we explored the transferability of PGS between EAS and EUR populations, which is critical information regarding the utility of PGS. Furthermore, our PGS analyses also indicated the advantages of ancestral diversity over a single population in PGS construction³⁷.

[Added to the Method, page 34, lines 797–798]

The details for the analyses under the same sample sizes are summarised in the **Supplementary Note**.

3. Lines 222-235: Some readers might not be impressed by the concordance between EAS and EUR at these top SNPs. Supplementary 12 makes it clear that the estimates in EAS are much more perturbed by sampling error. It might help to use a more formal framework incorporating variation in statistical power to derive benchmarks for comparison, such as the framework used by Okbay et al. (2022, SI Section 2.3)

Response: We thank reviewer for the valuable suggestion. In response to this suggestion, we have employed a formal framework to incorporate variation in statistical power. To achieve this, we have employed the approach proposed by Huang et al. (Ref 1), which aligns well with the nature of our cross-ancestry comparison. This method calculates the power-adjusted

transferability (PAT) ratio and utilises this ratio to assess the transferability of identified loci between different populations.

The result obtained from this approach is as follows:

No. of EAS	Loci associated in EUR	Observed transferable loci in EAS	Expected transferable loci in EAS	PAT ratio
176,400	246	95 (38.6%)	153 (62.2%)	0.62

The PAT ratio, obtained by dividing the number of observed transferable loci (95) by the number of expected transferable loci (153), resulted in a value of 0.62. This result indicates a relatively high transferability of GWAS loci identified in EUR populations to the EAS populations and is comparable to other phenotypes. For example, Huang et al. evaluated the transferability of GWAS loci for coronary artery disease between EUR and South Asian populations and reported a PAT ratio of 0.62. Additionally, Meng et al. (Ref 2) also employed this framework to assess the transferability of loci for major depression and reported a PAT ratio of 0.63 between the EUR and Hispanic/Latinx groups.

Ref 1. Huang, Qin Qin, et al. "Transferability of genetic loci and polygenic scores for cardiometabolic traits in British Pakistani and Bangladeshi individuals." *Nature communications* 13.1 (2022): 4664.

Ref 2. Meng, Xiangrui, et al. "Multi-ancestry GWAS of major depression aids locus discovery, fine-mapping, gene prioritisation, and causal inference." *bioRxiv* (2022): 2022-07.

[Added to the Results, pages 12–13, lines 263–271]

Assessment of transferability between EAS and EUR

We investigated the transferability of *EduYears* genomic loci identified in the EUR population to the EAS population with the power-adjusted transferability (PAT) ratio³³. To consider differences in LD patterns, we first generated credible sets for the 246 genetic loci associated with *EduYears* from Lee *et al.*¹⁵ study (n = 766,345). Based on the credible sets, the PAT ratio for *EduYears* for EUR to EAS was 0.62 (number of observed transferable loci divided by number of expected transferable loci in the EAS population = 95/153). This result indicates a relatively high transferability of GWAS loci associated with *EduYears* between EAS and EUR populations.

[Added to the Discussion, page 17, lines 382–386]

To facilitate cross-population comparisons, we investigated the transferability of *EduYears* loci between EAS and EUR populations using the PAT ratio approach³³, which considers the potential limitation of statistical power in the EAS population compared to EUR. Our findings indicate a relatively high transferability of *EduYears* loci identified in the EUR population to the EAS population.

[Added to the Method, pages 31–32, lines 726–740]

Assessment of transferability

To assess the transferability of *EduYears*-associated loci between EAS and EUR populations, we employed the PAT ratio approach³³. We initiated the analysis with 246 loci identified from publicly available EUR summary statistics (n=766,345) provided by Lee *et al.*¹⁵. For each locus, we generated credible sets, which included lead SNPs and proxy SNPs, using the same criteria as described in the study by Huang *et al.*³³. Specifically, we included SNPs within a 50 kb window

of the lead SNP with $r^2 \geq 0.8$ and $P\text{-value} < 100 \times P_{\text{lead}}$ using the 1KG Project phase 3 EUR data as the reference panel. A locus was considered "transferable" if at least one variant within its credible set exhibited an association with *EduYears* in the EAS population ($P\text{-value} < 0.05$) and demonstrated the same effect direction as observed in EUR. To estimate statistical power, we used the default parameter ($\alpha = 0.05$) and the summed-up power estimates for all published loci to obtain the expected number of transferable loci. Finally, by dividing the observed number of loci by the expected number of loci, we calculated the PAT ratio to estimate the transferability of *EduYears* loci between the EAS and EUR populations.

4. Lines 346-347: I suggest giving only the HUGO gene symbol for FAM81B rather than the full name that the symbol stands for.

Response: Thank you for the valuable comments. We have revised the sentence, as suggested.

5. Tables 1 and 2: What does "credible set ID" mean? What does it mean for a gene to be listed (or not) for a particular row? The table notes should explain these.

Response: Thanks for the valuable suggestion. Multiple credible sets were identified in certain regions, necessitating the use of a "Credible set ID" to distinguish between different credible sets within the same region. The "Gene" column displayed genes affected by the variant using the Variant Effect Predictor tool. In instances where no gene was affected by the variant, "NA" was indicated in the table. To provide enhanced clarity and context, we have incorporated several explanatory notes in **Tables 1** and **2** as follows:

“Credible set ID”: the ID of credible sets used to indicate different credible sets in the same region.

“Gene”: the genes affected by the variant using the Variant Effect Predictor tool.

“Annotation”: the consequence of variants on the protein sequence as annotated using the Variant Effect Predictor tool.

6. Supplementary Figure 5: This scatterplot is pretty useful. But even more useful might be a calculation of F_{ST} between the two EAS populations, either from the data in this study or in another publication. F_{ST} between the extremes of northern and southern Europe is about 0.01, and I believe it is common practice to include cohorts from all latitudes of Europe in EUR meta-analyses.

Response: Thank you for the valuable comments. We have added the distribution of fixation index (F_{st}) between TWB and KoGES in **Supplementary Table 4**, in which the mean F_{st} between TWB and KoGES was 0.005.

Supplementary Table 4. The distribution of F_{st} between TWB and KoGES

Item	Wright's F_{st}	Hudson's F_{st}
Minimum	0.0000	0.0000
1st quartile	0.0004	0.0004
Median	0.0020	0.0020
Mean	0.0046	0.0046
3rd quartile	0.0059	0.0059

Maximum	0.4375	0.4375
Ratio of average	0.0050	0.0050

7. Supplementary Tables 7-9: Why not include the LDSC-SEG enrichments as well as the coefficients?

Response: Thank you for the constructive suggestion. To obtain enrichment estimates for each annotation, we separately performed LDSC applied to Specifically Expressed Genes (LDSC-SEG) on the baseline annotations, along with one annotation from each gene set from Finucane et al. (Ref 1). As a result, the estimates of enrichment have been included, and the overall outcomes are now summarised in **Supplementary Tables 10-12**. Furthermore, there is an additional update in **Supplementary Table 12**, which shows the results of central nervous system (Cahoy) type of gene expression for *EduYears* in EAS using LDSC-SEG. For analyses using LDSC-based methods, we should have used EAS summary statistics obtained from PLINK. However, it was identified that summary statistics from Regenie had been used for this analysis. Therefore, we have updated **Supplementary Table 12** with the accurate result derived from PLINK summary statistics.

Ref 1. Finucane, Hilary K., et al. "Heritability enrichment of specifically expressed genes identifies disease-relevant tissues and cell types." *Nature genetics* 50.4 (2018): 621-629.

8. Supplementary Table 15: The source of the GWAS summary statistics is sometimes given as "Figshare." Is this a mistake?

Response: Thank you for pointing this out. We have recognized the inadequate provision of information in **Supplementary Table 15** (now **Supplementary Table 18**) regarding the sources of the GWAS summary statistics. We have revised the source information accordingly (now **Supplementary Table 18**).

We would like to express our gratitude for the valuable comments and suggestions. We really appreciate the time and effort invested in reviewing our manuscript.

Decision Letter, first revision:

13th October 2023

Dear Dr. Won,

Thank you for your patience as we've prepared the guidelines for final submission of your Nature Human Behaviour manuscript, "Shared genetic architectures of educational attainment in East Asian and European populations" (NATHUMBEHAV-23030871A). Please carefully follow the step-by-step instructions provided in the attached file, and add a response in each row of the table to indicate the changes that you have made. Please also address the additional marked-up edits we have proposed within the reporting summary. Ensuring that each point is addressed will help to ensure that your revised manuscript can be swiftly handed over to our production team.

We would hope to receive your revised paper, with all of the requested files and forms within two-three weeks. Please get in contact with us if you anticipate delays.

Nature Human Behaviour offers a Transparent Peer Review option for new original research manuscripts submitted after December 1st, 2019. As part of this initiative, we encourage our authors to support increased transparency into the peer review process by agreeing to have the reviewer comments, author rebuttal letters, and editorial decision letters published as a Supplementary item. When you submit your final files please clearly state in your cover letter whether or not you would like to participate in this initiative. Please note that failure to state your preference will result in delays in accepting your manuscript for publication.

In recognition of the time and expertise our reviewers provide to Nature Human Behaviour's editorial process, we would like to formally acknowledge their contribution to the external peer review of your manuscript entitled "Shared genetic architectures of educational attainment in East Asian and European populations". For those reviewers who give their assent, we will be publishing their names alongside the published article.

Cover suggestions

We welcome submissions of artwork for consideration for our cover. For more information, please see our https://www.nature.com/documents/Nature_covers_author_guide.pdf target="new"> guide for cover artwork.

ORCID

Non-corresponding authors do not have to link their ORCIDs but are encouraged to do so. Please note that it will not be possible to add/modify ORCIDs at proof. Thus, please let your co-authors know that if they wish to have their ORCID added to the paper they must follow the procedure described in the following link prior to acceptance:

Nature Human Behaviour has now transitioned to a unified Rights Collection system which will allow our Author Services team to quickly and easily collect the rights and permissions required to publish your work. Approximately 10 days after your paper is formally accepted, you will receive an email in providing

you with a link to complete the grant of rights. If your paper is eligible for Open Access, our Author Services team will also be in touch regarding any additional information that may be required to arrange payment for your article.

Please note that *Nature Human Behaviour* is a Transformative Journal (TJ). Authors may publish their research with us through the traditional subscription access route or make their paper immediately open access through payment of an article-processing charge (APC). Authors will not be required to make a final decision about access to their article until it has been accepted. Find out more about Transformative Journals

[REDACTED]

Best regards,
Alex McKay
Editorial Assistant
Nature Human Behaviour

On behalf of

Arunas Radzvilavicius, PhD
Senior Editor, Nature Human Behaviour
Nature Research

Reviewer #1:

Remarks to the Author:

Authors well responded to the reviewer's comments.

Reviewer #2:

Remarks to the Author:

I appreciate how responsive the authors have been to my previous comments and suggestions. I have no additional comments on the current version.

I sign my reviews

~Jaime Derringer

Reviewer #3:

Remarks to the Author:

In my first review I already stated that the paper was within an iota of being ready to be published and only had minor suggestions.

I have glanced over the revisions and see no reason to change my view. I do suggest that the authors remove the sentence "The field of behavioural genetics has progressed from a reductionist model ..."
The first parts sounds like a caricature of researchers in cognitive epidemiology.

Author Rebuttal, first revision:

Responses from the Authors to Review Comments:

We would like to express our gratitude for the valuable comments and suggestions. We really appreciate the time and effort invested in reviewing our manuscript.

REVIEWER COMMENTS:

Reviewer #1:

Remarks to the Author:

Authors well responded to the reviewer's comments.

Response: Thank you for valuable feedback and thoughtful suggestions and insights.

Reviewer #2:

Remarks to the Author:

I appreciate how responsive the authors have been to my previous comments and suggestions. I have no additional comments on the current version.

I sign my reviews

~Jaime Derringer

Response: Thank you for valuable feedback and thoughtful suggestions and insights.

Reviewer #3:

Remarks to the Author:

In my first review I already stated that the paper was within an iota of being ready to be published and only had minor suggestions.

I have glanced over the revisions and see no reason to change my view. I do suggest that the authors remove the sentence "The field of behavioural genetics has progressed from a

reductionist model ..." The first parts sounds like a caricature of researchers in cognitive epidemiology.

Response: We greatly appreciate this insightful comment from the reviewer. We agree with the reviewer's comment that we revised the sentence as follows:

As previous studies and our results suggest, *EduYears* shows phenotypic correlations and shares genetic components with multiple traits and diseases relevant to medical research, including cognitive function, neurodegeneration, and psychiatric disorders⁴⁴⁻⁴⁷, and findings on genetic overlaps between *EduYears* and health outcomes may shed light on the genetic basis of these relevant health outcomes. However, the link between *EduYears* and these health outcomes varies with context (such as nationality)^{48,49} and the impact of *EduYears* on health outcomes is likely via complex mechanisms like mediation and interaction between genetic and environmental factors⁴⁸. To this point, we would like to highlight that while understanding the genetic basis of *EduYears* (as a proxy phenotype) may improve our insights of other relevant health outcomes, our results do not support any immediate medical or clinical applications, such as polygenic prediction in direct-to-consumer services^{50,51}.

Final Decision Letter:

Dear Professor Won,

We are pleased to inform you that your Article "Shared genetic architectures of educational attainment in East Asian and European populations", has now been accepted for publication in *Nature Human Behaviour*.

Please note that *Nature Human Behaviour* is a Transformative Journal (TJ). Authors may publish their research with us through the traditional subscription access route or make their paper immediately open access through payment of an article-processing charge (APC). Authors will not be required to make a final decision about access to their article until it has been accepted. Find out more about Transformative Journals

With best regards,

Arunas Radzvilavicius, PhD
Senior Editor, Nature Human Behaviour
Nature Research